# Neutral-Reference Prompting for Vision–Language Models

Senmao Tian [1]  Xiang Wei [1]  Shunli Zhang [1]

## Abstract

Efficient transfer learning of vision–language models (VLMs) commonly suffers from a Base–New Trade-off (BNT): improving performance on unseen (new) classes often degrades accuracy on known (base) classes. Addressing how to boost recognition of unseen classes without sacrificing known-class performance remains a central challenge. Existing work often simplistically attributes the BNT to overfitting on known classes. We observe an interesting phenomenon: VLMs frequently exhibit asymmetric confusion on certain downstream data, i.e., samples of class A are systematically mispredicted as class B, while the reverse confusion (B → A) rarely occurs. For known classes, this kind of bias can be mitigated by tuning using a cross-entropy loss, but for unseen classes, such pretraining-induced bias persists and harms generalization. Motivated by this, we propose NeRP, a plug-and-play prompting correction strategy that improves discrimination on unseen classes without modifying model parameters. NeRP leverages neutral text prompts and reference images to measure class-wise prior preferences along the pre-trained inter-class geometry, and combines them with the sample likelihood to obtain the model's surrogate score. If, for a given sample, the prior strongly favors the current prediction while the observed evidence is clearly insufficient, we perform a local flip between easily confusable class pairs, thereby correcting prior-dominated mispredictions. Extensive experiments across multiple backbones and 15 few-shot and cross-domain benchmarks show that NeRP substantially improves accuracy on unseen classes while preserving known-class prediction performance.

[1]Beijing Jiaotong University, Beijing, China. Correspondence to: Shunli Zhang <smtian1204@gmail.com>.

*Proceedings of the $43^{rd}$ International Conference on Machine Learning*, Seoul, South Korea. PMLR 306, 2026. Copyright 2026 by the author(s).

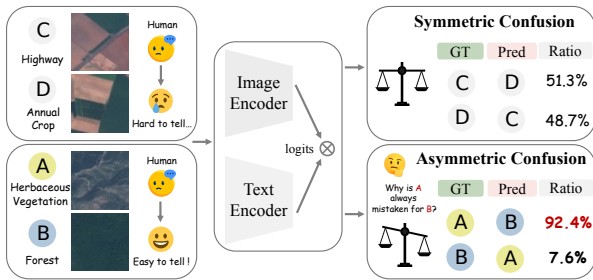

*Figure 1.* We observe two distinct types of confusion on downstream data. The first is symmetric confusion: for two similar classes the model struggles to discriminate them, samples from each class are misclassified as the other at comparable rates. The second is asymmetric confusion: the two classes may not be intrinsically ambiguous, yet the model systematically mislabels class A as B far more often than it mislabels B as A. More examples can also be found in Figure 5.

## 1. Introduction

Pre-trained VLMs (Radford et al., 2021; Jia et al., 2021; Alayrac et al., 2022; Yao et al., 2021; Huang et al., 2023; Peng et al., 2023; Lin et al., 2024) have demonstrated remarkable zero-shot capabilities in few-shot and cross-domain recognition tasks. However, prompt tuning and lightweight adaptation methods often suffer from the BNT problem (Zhang et al., 2024; Feng et al., 2026; Feng & Ge, 2025): improving performance on unseen (new) classes typically comes at the cost of degraded accuracy on seen (base) classes. Previous studies commonly attribute this phenomenon to overfitting (Li et al., 2025; Zhang et al., 2024; Khattak et al., 2023b; Li et al., 2024b; Yao et al., 2023; Liu et al., 2025; Pan et al., 2025) on base classes during fine-tuning, which compromises generalization to unseen classes. To mitigate overfitting, various methods have been explored, such as modifying loss functions (Khattak et al., 2023b; Yao et al., 2024), constraining prompt learning (Yao et al., 2024; Zhou et al., 2022a; Roy & Etemad, 2024; Lu et al., 2022), enhancing multimodal interaction (Yang et al., 2024; Khattak et al., 2023a; Guo & Gu, 2025), and incorporating external knowledge (Li et al., 2024b; Wu et al., 2024).

However, our investigation reveals that poor performance on unseen classes cannot be fully explained by overfitting alone. A more subtle issue is the asymmetric confusion as Figure 1 shows. This arises from data imbalance during VLM pre-training (Ghate et al., 2025; Wang et al., 2024;

Tu et al., 2023); therefore, on both the image side and the text side, there exists a certain preference for some classes. During fine-tuning, the cross-entropy loss helps the model learn correct class boundaries (Tian et al., 2026), effectively suppressing biases for seen classes. Nevertheless, for unseen classes, existing approaches (Zhou et al., 2022b; Gao et al., 2024; Zhu et al., 2023) heavily rely on the inherent zero-shot ability of vanilla VLMs. Consequently, the asymmetric biases inherited from pre-training become more pronounced on unseen classes, severely limiting generalization.

Identifying confusable class pairs and reversing the model's bias is appealing, but two issues remain: the preferred direction of confusion is unknown, and naïve flipping can harm correct predictions unless biased errors are detected with high confidence.

To address the above challenges, we first revisit how VLMs are efficiently adapted to downstream image classification (Radford et al., 2021; Li et al., 2021; Singh et al., 2022). Whether via prompt-based tuning or lightweight adapters, existing approaches essentially use learnable parameters to fit the statistics of the downstream data. These methods either fine-tune the text encoder or feed fine-tuned prompt vectors into a vanilla text encoder to produce a text embedding for each class; at inference, the image encoder embeds an unseen image (optionally with a learnable image prompt) and computes cosine similarities against all class embeddings to obtain the classification logits. For any image containing semantic content, the model naturally selects the text semantics closest to the image semantics as the predicted class. However, what should the model output for an image devoid of semantic information? This motivates our concept of *Neutral-Reference Prompt* (NeRP).

We define a *neutral-reference prompt* as one that can describe any sample in the dataset while remaining class-agnostic: a neutral-reference image preserves only domain-specific style with semantic content removed, and a neutral text prompt contains no class-specific cues. Our theoretical and empirical analyses show that fine-tuning mainly deforms a low-dimensional subspace spanned by prototypes of known classes, while the zero-shot geometry among unseen classes remains largely stable. Consequently, for any pair of unseen classes, the neutral prior gap induced by the neutral-reference prompt and the model's expected logit margin have the same sign with high probability. For an unbiased model, this gap is close to zero; otherwise, neutral-prior probing reveals the direction and strength of bias between confusable classes. We then construct Bayes-style posterior log-odds by adding this prior term to the observed sample logit margin. When the prior is strong but the evidence weak, we treat the prediction as prior-dominated and likely erroneous, and perform a local decision flip within the neighborhood of easily confusable classes; when both prior and

evidence are weak, we regard the model as genuinely undecided and leave the prediction unchanged. Extensive experiments show that NeRP integrates seamlessly with diverse state-of-the-art prompt tuning and lightweight adaptation methods, consistently improving unseen-class performance on nearly all downstream benchmarks without retraining or modifying model parameters, thereby serving as a plug-and-play prompting strategy that balances discrimination and generalization in VLMs.

## 2. Related Work

### 2.1. Efficient Transfer Learning for VLMs.

Vision-language models (VLMs) are designed to jointly understand images and text, with representative models like CLIP (Radford et al., 2021), ALIGN (Jia et al., 2021), and Flamingo (Alayrac et al., 2022) showcasing powerful cross-modal capabilities. To further enhance these models' performance on specific tasks or domains, various transfer learning methods have been proposed to adapt their pre-trained representations. Early work, such as CoOp (Zhou et al., 2022b) replaces hand-crafted templates with learnable context vectors while freezing CLIP. CoCoOp (Zhou et al., 2022a) further conditions prompts on each image to reduce base-to-new drift. ProGrad (Zhu et al., 2023) updates prompts only when gradients align with general knowledge, and DPC (Li et al., 2025) decouples optimization directions explicitly with a dual-prompt scheme. TCP (Yao et al., 2024) enhances discriminative capability by incorporating semantic priors through mapping class-level textual knowledge into class-specific prompt tokens. In a similar vein, PromptSRC (Khattak et al., 2023b) leverages tricks such as text diversification and self-ensembling to reduce overfitting, without incurring any inference-time cost. SkipT (Wu et al., 2025) makes VLMs self-adapters—skip low-impact layers and classes to fine-tune faster with no extra params.

To better align multimodal representations, MaPLe (Khattak et al., 2023a) couples deep prompts in both branches via a learned projection. Visual-side prompting has also been strengthened with progressive designs such as ProVP (Xu et al., 2024) to stabilize feature distributions. MMA (Yang et al., 2024) inserts multi-modal adapters and shows that higher layers carry task-specific signals while lower layers preserve generalization. MMRL (Guo & Gu, 2025) learns a shared representation space with extra tokens inserted only in higher layers, while regularizing the original class token to retain zero-shot priors, and uses a decoupled inference strategy.

### 2.2. Bias in Pretrained VLMs

Recent work has already studied biases in pretrained VLMs. Hamidieh *et al.* (Hamidieh et al., 2024) proposed the So-B-

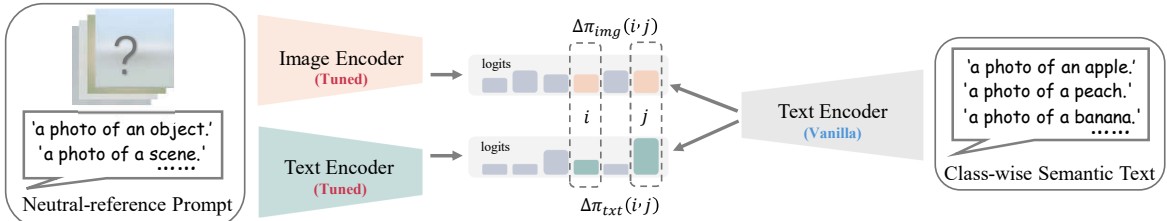

*Figure 2.* The pipeline of obtaining prior gaps from the neutral-reference prompt.

IT vocabulary and showed that CLIP over-associates harmful terms with certain demographics in open-vocabulary retrieval. Alabdulmohsin *et al.* (Alabdulmohsin et al., 2024) introduced Multi-Modal Moment Matching (M4) to balance data at both representation and association levels. Pushing further, Sahili *et al.* (Sahili et al., 2025) controlled model size, data scale, and data composition, and found that the "bigger and fairer" intuition does not generally hold. Ghate *et al.* (Ghate et al., 2025) quantified this more precisely: the choice of pretraining dataset is the dominant upstream factor. Beyond social bias, broader forms of bias have been explored. Wang *et al.* (Wang et al., 2024) built the CounterAnimal benchmark and revealed CLIP's reliance on spurious cues such as background. Tu *et al.* (Tu et al., 2023) revealed that CLIP's shape bias weakens after ImageNet-targeted fine-tuning while texture reliance increases. These insights also inform mitigation for downstream tasks. Li *et al.* (Li et al., 2024a) alleviate the overfitting caused by a single background class via learned background prompts and online mining. RS-CLIP (Li et al., 2023) uses structured adaptation and data cleaning to ease class-imbalance effects, transferring generic CLIP to RS scenes. Motivated by these findings, we develop a general, efficient approach for VLMs that suppresses bias in transfer learning.

## 3. Method

### 3.1. Preliminary

**Notation.** Let $\mathcal{X}$ be the image space and $\mathcal{C}$ the label set with $|\mathcal{C}| = C$. We write $d$ for the embedding dimension and assume L2-normalized features unless stated. We explicitly distinguish raw encoders from normalized features: let $g_{\mathrm{img}} : \mathcal{X} \to \mathbb{R}^d$ and $g_{\mathrm{txt}} : \mathrm{Token}^* \to \mathbb{R}^d$ denote *raw image* and text encoders, and define the normalized outputs

$$f_{\mathrm{img}}(x) := \mathrm{norm}\big(g_{\mathrm{img}}(x)\big), f_{\mathrm{txt}}(z) := \mathrm{norm}\big(g_{\mathrm{txt}}(z)\big).$$

Let $g_{\mathrm{img}}^0, g_{\mathrm{txt}}^0$ be the corresponding *pretrained (zero-shot)* raw encoders, and $f_{\mathrm{img}}^0 := \mathrm{norm} \circ g_{\mathrm{img}}^0, f_{\mathrm{txt}}^0 := \mathrm{norm} \circ g_{\mathrm{txt}}^0$ their normalized counterparts (CLIP-like). For each class $c \in \mathcal{C}$ we define the class text prototypes

$$t(c) := \mathrm{norm}\big(g_{\mathrm{txt}}(\mathrm{prompt}(c))\big),$$
$$t^0(c) := \mathrm{norm}\big(g_{\mathrm{txt}}^0(\mathrm{prompt}_0(c))\big).$$

Given an image $x \in \mathcal{X}$, the class score is $\ell_c(x) := \langle f_{\mathrm{img}}(x), t(c) \rangle$. We split the label set into base $\mathcal{B}$ (seen during fine-tuning) and novel $\mathcal{N} = \mathcal{C} \setminus \mathcal{B}$ (unseen). Throughout, we fix the *base subspace*

$$S := \mathrm{span}\{ t^0(c) : c \in \mathcal{B} \} \subset \mathbb{R}^d.$$

For a dataset/domain $D$, let $\tau(D)$ denote the dataset-specific *neutral-reference text prompt* (e.g., "a photo of an object.").

**Neutral priors.** For the text side, we construct a zero-shot neutral vector

$$u_{\mathrm{txt}}^0(D) := \mathrm{norm}\big(g_{\mathrm{txt}}^0(\tau(D))\big).$$

The per-class *text prior logit* is

$$\pi_{\mathrm{txt}}(c; D) := \big\langle t(c), u_{\mathrm{txt}}^0(D) \big\rangle, c \in \mathcal{C}. \tag{1}$$

For the image side, we use the feature of the mean image (pixel-wise average of the preprocessed training images) as the *neutral-reference image prompt* and then normalize:

$$\bar{x}^D := \frac{1}{n} \sum_{x \in \mathrm{train}(D)} \mathrm{preproc}(x),$$
$$u_{\mathrm{img}}(D) := f_{\mathrm{img}}(\bar{x}^D) = \mathrm{norm}\big(g_{\mathrm{img}}(\bar{x}^D)\big).$$

Correlating with *zero-shot* class prototypes yields the per-class *image prior logit*

$$\pi_{\mathrm{img}}(c; D) := \big\langle u_{\mathrm{img}}(D), t^0(c) \big\rangle, c \in \mathcal{C}. \tag{2}$$

For a class pair $(i, j)$ we define the *prior gaps*

$$\Delta\pi_{\mathrm{txt}}(i, j) := \pi_{\mathrm{txt}}(i; D) - \pi_{\mathrm{txt}}(j; D),$$
$$\Delta\pi_{\mathrm{img}}(i, j) := \pi_{\mathrm{img}}(i; D) - \pi_{\mathrm{img}}(j; D).$$

An intuitive illustration is shown in Figure 2.

### 3.2. When & Why Priors are Informative

**In a nutshell.** (i) Fine-tuning mainly reshapes a low-dimensional "base" subspace spanned by base class prototypes; (ii) zero-shot geometry between classes is relatively stable across target domains; (iii) fine-tuning on base classes warps the pretrained inter-class directions, so our neutral priors become unreliable on base classes but remain informative for novel classes.

**Preparation.** We model fine-tuning as a low-rank deformation (Hu et al., 2022; He et al., 2021; Wang et al., 2023) of the *raw* pretrained geometry:

$$g_{\text{img}}(x) = g^0_{\text{img}}(x) + U_{\text{img}}\, a(x),$$
$$g_{\text{txt}}(z) = g^0_{\text{txt}}(z) + U_{\text{txt}}\, b(z),$$

(3)

with $U_{\text{img}}, U_{\text{txt}} \in \mathbb{R}^{d \times r}$, $r \ll d$, and coefficient maps $a(\cdot), b(\cdot)$ that are *large* on $\mathcal{B}$ and *small on average* on $\mathcal{N}$. Normalized features are obtained by $f = \text{norm}(g)$ and $t(\cdot) = \text{norm}(g_{\text{txt}}(\cdot))$. We define the pretrained inter-class direction $\Delta^0_{ij} := t^0(i) - t^0(j)$ and, for any feature $v \in \mathbb{R}^d$, the (signed) separation $\delta^0_{ij}(v) := \langle v, \Delta^0_{ij} \rangle$. We write $\Pi_T$ for the Euclidean orthogonal projector onto a subspace $T$.

**Assumption 3.1** (Base subspace). Recall $S = \text{span}\{t^0(c) : c \in \mathcal{B}\}$. For most $(i, j) \in \mathcal{B} \times \mathcal{B}$, $\|\Pi_{S^\perp}(t(i) - t(j))\| \ll \|\Pi_S(t(i) - t(j))\|$.

In practice, efficient fine-tuning of VLMs—using only a small number of trainable parameters or low-rank adapters—often matches full fine-tuning on the base classes $\mathcal{B}$, indicating that discriminative updates concentrate along a few shared directions (i.e., within $(S)$).

**Assumption 3.2** (Zero-shot stability). For $x \sim D$ labeled in $\mathcal{N}$ and typical confusable pairs $(i, j) \in \binom{\mathcal{N}}{2}$,

$$\text{Var}_{x \sim D}\big[\delta^0_{ij}(f^0_{\text{img}}(x))\big] \leq \sigma^2,$$

with a small, domain-insensitive $\sigma^2$.

The assumption follows from a one-line Rayleigh-quotient argument under mild conditions, for example: let $\|\Delta^0_{ij}\| = 1$, and define $\Omega_D := \text{Cov}_{x \sim D}[f^0_{\text{img}}(x)]$, then

$$\text{Var}_{x \sim D}\big[\delta^0_{ij}(f^0_{\text{img}}(x))\big] = (\Delta^0_{ij})^\top \Omega_D \Delta^0_{ij} \leq \lambda_{\max}(\Omega_D),$$

where $\lambda_{\max}(\Omega_D)$ denotes the largest eigenvalue of $\Omega_D$. Hence, when specific conditions are met, Assumption 3.2 holds as a lemma. This assumption is also consistent with modeling L2-normalized embeddings on the unit sphere via a von Mises–Fisher family (Lu et al., 2024; Banerjee et al., 2005).

**Lemma 3.3** (Gaps as projections). *For $(i, j) \in \mathcal{N} \times \mathcal{N}$,*

$$\Delta\pi_{\text{img}}(i, j) = \langle u_{\text{img}}(D), \Delta^0_{ij} \rangle,$$
$$\Delta\pi_{\text{txt}}(i, j) = \langle \Delta^0_{ij}, u^0_{\text{txt}}(D) \rangle$$
$$+ \langle U_{\text{txt}}\big(b(i) - b(j)\big), u^0_{\text{txt}}(D) \rangle.$$

*In particular,*

$$\big|\Delta\pi_{\text{txt}}(i, j) - \langle \Delta^0_{ij}, u^0_{\text{txt}}(D) \rangle\big|$$
$$\lesssim \|U_{\text{txt}}\|\big(\|b(i)\| + \|b(j)\|\big),$$

$\lesssim$ *denotes inequality up to an absolute constant factor. So both priors probe the same pretrained direction $\Delta^0_{ij}$ up to a small novel-class residual.*

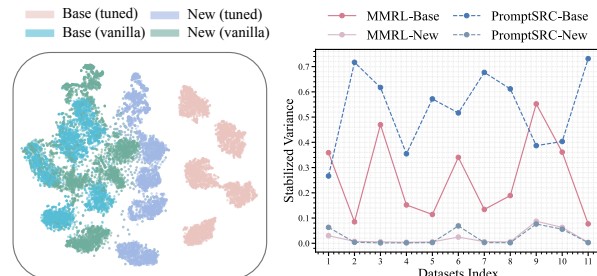

*Figure 3.* The t-SNE visualization of embeddings, and the comparison of between-class variance of per-class mean logits.

*Takeaway.* The image prior uses a mean-image feature anchor; the text prior uses a neutral text anchor. Both are linear "rulers" aligned with pretrained inter-class directions on novel classes.

**Lemma 3.4** (Base suppression). *Conditional on Assumption 3.1, if $\|\Pi_S u_{\text{img}}(D)\| \leq \kappa_{\text{img}}$ and $\|\Pi_S u^0_{\text{txt}}(D)\| \leq \kappa_{\text{txt}}$, then for any $(i, j) \in \mathcal{B} \times \mathcal{B}$,*

$$|\Delta\pi_{\text{img}}(i, j)| \leq \kappa_{\text{img}}\|\Pi_S \Delta^0_{ij}\|,$$
$$|\Delta\pi_{\text{txt}}(i, j)| \leq \kappa_{\text{txt}}\|\Pi_S(t(i) - t(j))\|$$
$$+ \|\Pi_{S^\perp} u^0_{\text{txt}}(D)\| \|\Pi_{S^\perp}(t(i) - t(j))\|.$$

*Takeaway.* Because base directions live in $S$ and the neutral anchors have little energy in $S$, neutral-prior gaps are intrinsically small on $\mathcal{B}$—yet remain informative off $S$ for novel classes.

We have also visualized the spatial distribution of embeddings on base and new classes using t-SNE, as well as the stabilized variance of the output logits with neutral-reference images as input on different datasets as Figure 3 shows.

For a pair $(i, j)$, define the logits margin:

$$m_{ij}(x) := \ell_i(x) - \ell_j(x) = \langle f_{\text{img}}(x), t(i) - t(j) \rangle,$$

and its expectation $\mu_{ij}(D) := \mathbb{E}_{x \sim D}[m_{ij}(x)]$. By combining Assumption 3.1, Assumption 3.2, Lemma 3.3 and Lemma 3.4, we can explain why priors are informative.

**Proposition 3.5** (Sign consistency). *Under the low-rank deformation model in (3) and Assumption 3.2, there exist domain- and model-dependent constants*

$$\varepsilon^{\mathcal{N}}_{\text{img}}, \ \varepsilon^{\mathcal{N}}_{\text{txt}} \ \text{(small on } \mathcal{N}\text{)}$$

*such that for any $(i, j) \in \mathcal{N} \times \mathcal{N}$,*

$$\big|\mu_{ij}(D) - \Delta\pi_{\text{img}}(i, j)\big| \leq \varepsilon^{\mathcal{N}}_{\text{img}},$$
$$\big|\mu_{ij}(D) - \Delta\pi_{\text{txt}}(i, j)\big| \leq \varepsilon^{\mathcal{N}}_{\text{txt}}.$$

*Consequently,* $\text{sign } \mu_{ij}(D) = \text{sign } \Delta\pi_{\text{img}}(i, j)$ *and* $\text{sign } \mu_{ij}(D) = \text{sign } \Delta\pi_{\text{txt}}(i, j)$ *whenever the gaps exceed the respective $\varepsilon$.*

On novel classes, because the inter-class direction satisfies $t(i) - t(j) \approx \Delta_{ij}^0$ and the low-rank deformation implies $f_{\mathrm{img}}(x) - f_{\mathrm{img}}^0(x)$ is small on $\mathcal{N}$, thus we have $\mathrm{Var}[m_{ij}(x)] \leq \sigma^2 + o(1)$, i.e., $\sigma_m^2 \lesssim \sigma^2$.

**Corollary 3.6** (High-probability sign consistency)**.** *Let* $(i,j) \in \mathcal{N} \times \mathcal{N}$ *and write* $\mu := \mu_{ij}(D)$ *and* $\sigma_m^2 := \mathrm{Var}_{x \sim D}[m_{ij}(x)]$*. Moreover, on* $\mathcal{N}$ *we have* $\sigma_m^2 \lesssim \sigma^2$*. Then for any* $\gamma > 0$,

$$\mathbb{P}_{x \sim D}\Big[ \mathrm{sign}\, m_{ij}(x) \neq \mathrm{sign}\, \mu \Big] \leq \min\left\{ \frac{\sigma_m^2}{(|\mu| - \gamma)^2}, 1 \right\},$$

*whenever* $|\mu| > \gamma$*. In particular, if* $|\Delta\pi_{\mathrm{img}}(i,j)| > \varepsilon_{\mathrm{img}}^{\mathcal{N}} + \gamma$ *(or* $|\Delta\pi_{\mathrm{txt}}(i,j)| > \varepsilon_{\mathrm{txt}}^{\mathcal{N}} + \gamma$*), then with probability at least* $1 - \sigma_m^2 / (|\Delta\pi| - \varepsilon - \gamma)^2$ *the sample-level sign agrees with the prior.*

*Interpretation.* For novel classes, both priors measure the zero-shot inter-class bias direction using neutral anchors. Low-rank fine-tuning perturbs this geometry weakly on $\mathcal{N}$, so the *expected* logits gap aligns with the prior gap up to a small error; the zero-shot stability assumption then upgrades this to a *high-probability* statement at the sample level.

Formal proof and more details are in the Appendix.

### 3.3. Bayes-Style Surrogate

**Neutral-prior composition.** For a domain $D$ and class $c \in \mathcal{C}$, recall

$$\pi_{\mathrm{txt}}(c; D) = \langle t(c),\, u_{\mathrm{txt}}^0(D) \rangle.$$

On the image side, recall $u_{\mathrm{img}}(D) := f_{\mathrm{img}}(\bar{x}^D)$ and

$$\pi_{\mathrm{img}}(c; D) = \langle u_{\mathrm{img}}(D),\, t^0(c) \rangle.$$

Given a pair $(i,j)$, we form a composite neutral prior gap

$$\Sigma_{i,j}(D) := \big(\pi_{\mathrm{txt}}(i; D) - \pi_{\mathrm{txt}}(j; D)\big) \\ + \big(\pi_{\mathrm{img}}(i; D) - \pi_{\mathrm{img}}(j; D)\big). \quad (4)$$

Note $\Sigma_{i,j}(D) = -\Sigma_{j,i}(D)$. We restrict comparisons to a symmetric neighborhood graph $G \subset \mathcal{C} \times \mathcal{C}$ (i.e., $(i,j) \in G \Leftrightarrow (j,i) \in G$) and write $\mathcal{A}(i) = \{j \neq i : (i,j) \in G\}$. To absorb domain-wide constant bias caused by fine-tuning in the composite neutral prior, we use a global intercept on the base pairs:

$$\hat{\beta}(D) := \underset{\beta}{\arg\min} \sum_{(i,j) \in \mathcal{B} \times \mathcal{B}} \big(\hat{\mu}_{ij}(D) - \Sigma_{i,j}(D) - \beta\big)^2, \quad (5)$$

whose closed-form solution $\beta^\star(D)$ and $\hat{\mu}_{ij}(D)$ is the empirical estimated domain mean of $m_{ij}(x)$, details of both are given in the Appendix. To reduce complexity, we combine it with $\tau$ in Equation (11) in practice.

**Residual form for priors.** Considering the diverse styles of downstream data, we provide a residual form to better adapt to semantically diverse datasets (e.g., ImageNet). On such datasets, inter-class semantics and text prototypes are more widely separated; combining Assumption 3.2, the across-pair variance of zero-shot projections (and thus logits gaps) becomes larger. To stabilize the neutral priors and cancel common, domain/prompt–induced bias, we *residualize* the image- or text-side prior by subtracting the correlation with the *current* anchor from that with the *zero-shot* anchor—thereby measuring correlation with the anchor displacement and retaining only domain-induced changes.

Let the current-model text anchor under the neutral prompt $\tau(D)$ be $u_{\mathrm{txt}}(D) := \mathrm{norm}\big(g_{\mathrm{txt}}(\tau(D))\big)$. We define the *text residual prior*

$$\tilde{\pi}_{\mathrm{txt}}(c; D) := \langle t(c),\, u_{\mathrm{txt}}^0(D) \rangle - \langle t(c),\, u_{\mathrm{txt}}(D) \rangle. \quad (6)$$

Similarly, using $u_{\mathrm{img}}(D)$ and class prototypes, we define the *image residual prior*

$$\tilde{\pi}_{\mathrm{img}}(c; D) := \langle u_{\mathrm{img}}(D),\, t^0(c) \rangle - \langle u_{\mathrm{img}}(D),\, t(c) \rangle. \quad (7)$$

The pairwise gaps become

$$\Delta\tilde{\pi}_{\mathrm{txt}}(i,j) = \Delta\pi_{\mathrm{txt}}(i,j) - \langle t(i) - t(j), u_{\mathrm{txt}}(D) \rangle,$$
$$\Delta\tilde{\pi}_{\mathrm{img}}(i,j) = \Delta\pi_{\mathrm{img}}(i,j) - \langle u_{\mathrm{img}}(D), t(i) - t(j) \rangle,$$

Intuitively, $\Delta\tilde{\pi}_{\mathrm{txt}}(i,j) \approx \langle \Delta_{ij}^0, u_{\mathrm{txt}}^0(D) - u_{\mathrm{txt}}(D) \rangle$ projects the pretrained inter-class direction onto the anchor displacement, filtering out class-independent bias and reducing across-pair variance on semantically diverse datasets; $\Delta\tilde{\pi}_{\mathrm{img}}(i,j)$ analogously measures deformation of inter-class directions and is typically small on novel classes.

We then use

$$\tilde{\Sigma}_{i,j}(D) := \big(\tilde{\pi}_{\mathrm{txt}}(i; D) - \tilde{\pi}_{\mathrm{txt}}(j; D)\big) \\ + \big(\tilde{\pi}_{\mathrm{img}}(i; D) - \tilde{\pi}_{\mathrm{img}}(j; D)\big) \quad (8)$$

as a drop-in replacement for $\Sigma_{i,j}(D)$ in (4). This replacement preserves the sign-consistency guarantee (Proposition 3.5) and the high-probability of this consistency (Corollary 3.6), often with tighter constants due to variance reduction; full details are deferred to the Appendix.

**Prior-dominated inconsistency.** For a test image $x$ with top-1 prediction $\hat{y}(x) = i$, and any $j \in \mathcal{A}(i)$, recall the sample-level margin $m_{ij}(x) = \ell_i(x) - \ell_j(x)$. We interpret $\Sigma_{i,j}(D)$ as a prior logit-style offset for $i$ against $j$, and $m_{ij}(x)$ as the data-dependent logit margin. $m_{ij}(x)$ comes from L2-normalized cosine similarities between embeddings, and under a von Mises–Fisher model, this margin can be approximately interpreted as a log-likelihood ratio. A Bayes-inspired surrogate score is then

$$s_{ij}(x) \approx m_{ij}(x) + \Sigma_{i,j}(D) + \hat{\beta}(D). \quad (9)$$

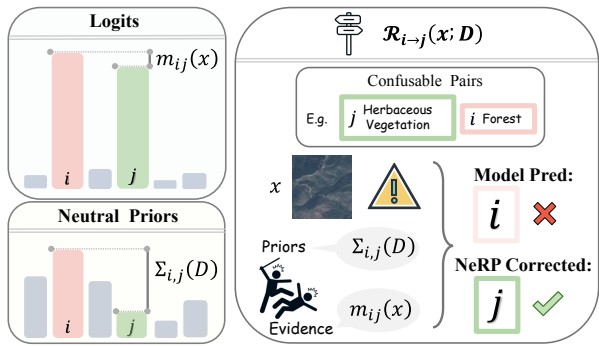

*Figure 4.* An explanation of how NeRP corrects mispredictions.

By Proposition 3.5 and additivity of errors, on novel classes there exists $\varepsilon^{\mathcal{N}} = \varepsilon_{\text{img}}^{\mathcal{N}} + \varepsilon_{\text{txt}}^{\mathcal{N}}$ such that

$$\left| \mu_{ij}(D) - \Sigma_{i,j}(D) \right| \leq \varepsilon^{\mathcal{N}}. \tag{10}$$

Thus, if the prior strongly favors $i$ against $j$ (i.e., $\Sigma_{i,j}(D)$ is large), we want the *expected* margin $\mu_{ij}(D)$ to be comparably large as well. When, however, the *observed* margin $m_{ij}(x)$ is small, the posterior surrogate in (9) is driven mainly by the prior term rather than by sample evidence. We then declare a prior-dominated region by prior gate $\tau$ and evidence gate $\delta$:

$$\mathcal{R}_{i \to j}(x; D) := \left\{ \Sigma_{i,j}(D) \geq \tau - \hat{\beta}(D) \wedge m_{ij}(x) \leq \delta \right\}. \tag{11}$$

Inside $\mathcal{R}_{i \to j}$, the gap $\Sigma_{i,j}(D) - m_{ij}(x) \geq \tau - \delta - \hat{\beta}(D)$ indicates the surrogate posterior is predominantly prior-driven.

**Decision rule.** As Figure 4 shows, for each $j \in \mathcal{A}(i)$, keep $j$ iff $(x, D) \in \mathcal{R}_{i \to j}$. Among retained neighbors, select

$$j^\star \in \arg \max_{j : (x, D) \in \mathcal{R}_{i \to j}} \ell_j(x),$$

and flip by enforcing a minimal tie-break on logits, e.g.,

$$\ell_{j^\star}(x) \leftarrow \max\{\ell_{j^\star}(x), \ell_i(x) + \varepsilon_0\} \quad (\varepsilon_0 > 0).$$

If no neighbor passes, leave logits unchanged.

**Risk–Benefit Analysis.** Recall $\beta^\star(D)$ from Equation (5), and denote $\Delta_\beta := \hat{\beta}(D) - \beta^\star(D)$. By Prop. 3.5 and error additivity, there exists

$$\varepsilon_{\text{int}}^{\mathcal{N}} := \varepsilon^{\mathcal{N}} + |\Delta_\beta|,$$
$$\text{s.t.} \quad \left| \mu_{ij}(D) - \left( \Sigma_{i,j}(D) + \hat{\beta}(D) \right) \right| \leq \varepsilon_{\text{int}}^{\mathcal{N}}. \tag{12}$$

Define the *effective gate* $\tilde{\tau} := \tau - \hat{\beta}(D)$, and let $\sigma_m^2 := \text{Var}_{x \sim D}[m_{ij}(x)] \lesssim \sigma^2$ under Assumption 3.2. On the event $\{\Sigma_{i,j} \geq \tilde{\tau}\}$, Equation (12) implies $\mu_{ij}(D) \geq \tau - \varepsilon_{\text{int}}^{\mathcal{N}}$. For any $\delta$ with

$$\gamma := \tau - \varepsilon_{\text{int}}^{\mathcal{N}} - \delta > 0,$$

Chebyshev yields

$$\mathbb{P}_{x \sim D}\left[ m_{ij}(x) \leq \delta \right] \leq \frac{\sigma_m^2}{\gamma^2} = \frac{\sigma_m^2}{\left( \tau - \varepsilon_{\text{int}}^{\mathcal{N}} - \delta \right)^2}.$$

Because a flip can only occur when $m_{ij}(x) \leq \delta$, the above equation upper-bounds the *false-flip* probability. Importantly, this requires only $\tau > \varepsilon_{\text{int}}^{\mathcal{N}} + \delta$. In practice, we do not explicitly look for $\tau$ but directly seek a suitable $\tilde{\tau}$.

## 4. Experiments

### 4.1. Settings

**Datasets & Baselines** Following previous works (Zhou et al., 2022b), we evaluate base-to-new generalization and cross-dataset transfer on eleven recognition benchmarks spanning diverse data distributions: ImageNet (Deng et al., 2009), Caltech101 (Fei-Fei et al., 2004), OxfordPets (Parkhi et al., 2012), StanfordCars (Krause et al., 2013), Flowers102 (Nilsback & Zisserman, 2008), Food101 (Bossard et al., 2014), FGVCAircraft (Maji et al., 2013), SUN397 (Xiao et al., 2010), DTD (Cimpoi et al., 2014), EuroSAT (Helber et al., 2019), and UCF101 (Soomro et al., 2012). For cross-domain evaluation, we consider ImageNet-V2 (Recht et al., 2019), ImageNet-Sketch (Wang et al., 2019), ImageNet-A (Hendrycks et al., 2021b), and ImageNet-R (Hendrycks et al., 2021a). All experiments utilize a 16-shot setting (16 training samples per class), and accuracy is the average of 3 random seeds. We select 4 influential prompt learners (CoCoOp, PromptSRC, MMA and MMRL) with different styles as baselines for our plug-and-play strategy. We also include several state-of-the-art (SOTA) methods as reference baselines.

**Confusable Pairs Generation** Let $\mathcal{C}$ denote the label set. For each class $i \in \mathcal{C}$, we query a locally deployed Qwen2.5-72B-Instruct (Yang et al., 2025) with the class name and description to produce at most $K$ candidate classes that are likely to be confused with $i$ (where $K$ scales with $|\mathcal{C}|$). Aggregating the resulting class pairs over all $i$, we construct an undirected adjacency graph $G$ on $\mathcal{C}$; an edge $(i, j)$ indicates that $j$ is a confusable neighbor of $i$. The prompt template is provided in the appendix.

**Target-Emulated Calibration** NeRP's decisions rely on two gates: the Prior Effective Gate $\tilde{\tau}$ and the Evidence Strength Gate $\delta$. Because the test data are unseen, we construct simulated subsets from the training data to select appropriate $\tilde{\tau}$ and $\delta$ for each task.

For the **base-to-novel** setting, we partition the base classes into $e$ disjoint folds $\{F_e^{b2n}\}_{e=1}^E$. In fold $e$, the pseudo-base classes are $\mathcal{B}_e = \mathcal{B} \setminus F_e$ and the pseudo-novel classes are $\mathcal{N}_e = F_e$. For the **cross-dataset setting**, following prior work, we group ImageNet-1K into superclasses, split them into $e$ disjoint subsets $\{F_e^{cd}\}_{e=1}^E$, and assign pseudo-source

*Table 1.* Base-to-novel generalization experiments of five baselines on 11 datasets. "•" denotes keeping the original output.

| Method | Average | | | ImageNet | | | Caltech101 | | | OxfordPets | | |
|---|---|---|---|---|---|---|---|---|---|---|---|---|
| | Base | Novel | HM | Base | Novel | HM | Base | Novel | HM | Base | Novel | HM |
| CoOp (IJCV 22) | 82.69 | 63.22 | 71.66 | 76.47 | 67.88 | 71.92 | 98.00 | 89.81 | 93.73 | 93.67 | 95.29 | 94.47 |
| MaPLe (CVPR 23) | 82.28 | 75.14 | 78.55 | 76.66 | 70.54 | 73.47 | 97.74 | 94.36 | 96.02 | 95.43 | 97.76 | 96.58 |
| CoPrompt (ICLR 24) | 84.00 | 77.23 | 80.48 | 77.67 | 71.27 | 74.33 | 98.27 | 94.90 | 96.55 | 95.67 | 98.10 | 96.87 |
| SkipT (CVPR 25) | 85.04 | 77.53 | 81.11 | 77.73 | 70.40 | 73.89 | 98.50 | 95.33 | 96.89 | 95.70 | 97.87 | 96.77 |
| CoCoOp (CVPR 22) | 80.47 | 71.69 | 75.83 | 75.98 | 70.43 | 73.10 | 97.96 | 93.81 | 95.84 | 95.20 | 97.69 | 96.43 |
| +NeRP (Ours) | • | **72.99** | **76.67** | • | 71.37 | 73.60 | • | 94.07 | 95.98 | • | 97.90 | 96.53 |
| PromptSRC (ICCV 23) | 84.26 | 76.10 | 79.97 | 77.60 | 70.73 | 74.01 | 98.10 | 94.03 | 96.02 | 95.33 | 97.30 | 96.30 |
| +NeRP (Ours) | • | **76.87** | **80.40** | • | 71.33 | 74.33 | • | 94.47 | 96.25 | • | 97.40 | 96.35 |
| MMA (CVPR 24) | 83.20 | 76.80 | 79.87 | 77.31 | 71.00 | 74.02 | 98.40 | 94.00 | 96.15 | 95.40 | 98.07 | 96.72 |
| +NeRP (Ours) | • | **77.59** | **80.30** | • | 71.73 | 74.42 | • | 94.27 | 96.29 | • | 98.13 | 96.75 |
| MMRL (CVPR 25) | 85.68 | 77.16 | 81.20 | 77.90 | 71.30 | 74.45 | 98.97 | 94.50 | 96.68 | 95.90 | 97.60 | 96.74 |
| +NeRP (Ours) | • | **78.04** | **81.68** | • | 71.53 | 74.58 | • | 95.10 | 97.00 | • | 97.70 | 96.79 |

| Method | StanfordCars | | | Flowers102 | | | Food101 | | | FGVCAircraft | | |
|---|---|---|---|---|---|---|---|---|---|---|---|---|
| | Base | Novel | HM | Base | Novel | HM | Base | Novel | HM | Base | Novel | HM |
| CoOp (IJCV 22) | 78.12 | 60.40 | 68.13 | 97.60 | 59.67 | 74.06 | 88.33 | 82.26 | 85.19 | 40.44 | 22.30 | 28.75 |
| MaPLe (CVPR 23) | 72.94 | 74.00 | 73.47 | 95.92 | 72.46 | 82.56 | 90.71 | 92.05 | 91.38 | 37.44 | 35.61 | 36.50 |
| CoPrompt (ICLR 24) | 76.97 | 74.40 | 75.66 | 97.27 | 76.60 | 85.71 | 90.73 | 92.07 | 91.40 | 40.20 | 39.33 | 39.76 |
| SkipT (CVPR 25) | 82.93 | 72.50 | 77.37 | 98.57 | 75.80 | 85.70 | 90.67 | 92.03 | 91.34 | 45.37 | 37.13 | 40.84 |
| CoCoOp (CVPR 22) | 70.49 | 73.59 | 72.01 | 94.87 | 71.75 | 81.71 | 90.70 | 91.29 | 90.99 | 33.41 | 23.71 | 27.74 |
| +NeRP (Ours) | • | 74.13 | 72.26 | • | 72.03 | 81.89 | • | 91.33 | 91.01 | • | 26.83 | 29.76 |
| PromptSRC (ICCV 23) | 78.27 | 74.97 | 76.58 | 98.07 | 76.50 | 85.95 | 90.67 | 91.53 | 91.10 | 42.73 | 37.87 | 40.15 |
| +NeRP (Ours) | • | 75.70 | 76.96 | • | 77.37 | 86.50 | • | 91.53 | 91.10 | • | 38.07 | 40.27 |
| MMA (CVPR 24) | 78.50 | 73.10 | 75.70 | 97.77 | 75.93 | 85.48 | 90.13 | 91.30 | 90.71 | 40.57 | 36.33 | 38.33 |
| +NeRP (Ours) | • | 73.97 | 76.17 | • | 77.03 | 96.17 | • | 91.30 | 90.71 | • | 36.63 | 38.50 |
| MMRL (CVPR 25) | 81.30 | 75.07 | 78.06 | 98.97 | 77.27 | 86.78 | 90.57 | 91.50 | 91.03 | 46.30 | 37.03 | 41.15 |
| +NeRP (Ours) | • | 75.30 | 78.19 | • | 78.43 | 87.51 | • | 91.55 | 91.06 | • | 37.67 | 41.54 |

| Method | SUN397 | | | DTD | | | EuroSAT | | | UCF101 | | |
|---|---|---|---|---|---|---|---|---|---|---|---|---|
| | Base | Novel | HM | Base | Novel | HM | Base | Novel | HM | Base | Novel | HM |
| CoOp (IJCV 22) | 80.60 | 65.89 | 72.51 | 79.44 | 41.18 | 54.24 | 92.19 | 54.74 | 68.69 | 84.69 | 56.05 | 67.46 |
| MaPLe (CVPR 23) | 80.82 | 78.70 | 79.75 | 80.36 | 59.18 | 68.16 | 94.07 | 73.23 | 82.35 | 83.00 | 78.66 | 80.77 |
| CoPrompt (ICLR 24) | 82.63 | 80.03 | 81.30 | 83.13 | 64.73 | 72.79 | 94.60 | 78.57 | 85.84 | 86.90 | 79.57 | 83.07 |
| SkipT (CVPR 25) | 82.40 | 79.03 | 80.68 | 83.77 | 67.23 | 74.59 | 92.47 | 83.00 | 87.48 | 87.30 | 82.47 | 84.81 |
| CoCoOp (CVPR 22) | 79.74 | 76.86 | 78.27 | 77.01 | 56.00 | 64.85 | 87.49 | 60.04 | 71.21 | 82.33 | 73.45 | 77.64 |
| +NeRP (Ours) | • | 77.46 | 78.58 | • | 56.67 | 65.29 | • | 67.30 | 76.08 | • | 73.77 | 77.82 |
| PromptSRC (ICCV 23) | 82.67 | 78.47 | 80.52 | 83.37 | 62.97 | 71.75 | 92.90 | 73.90 | 82.32 | 87.10 | 78.80 | 82.74 |
| +NeRP (Ours) | • | 78.90 | 80.74 | • | 63.37 | 72.01 | • | 78.53 | 85.11 | • | 78.93 | 82.81 |
| MMA (CVPR 24) | 82.27 | 78.57 | 80.38 | 83.20 | 65.63 | 73.38 | 85.46 | 82.34 | 83.87 | 86.23 | 80.03 | 82.20 |
| +NeRP (Ours) | • | 78.77 | 80.48 | • | 66.20 | 73.73 | • | 85.27 | 85.36 | • | 80.17 | 83.09 |
| MMRL (CVPR 25) | 83.20 | 79.30 | 81.20 | 85.67 | 65.00 | 73.82 | 95.60 | 80.17 | 87.21 | 88.10 | 80.07 | 83.89 |
| +NeRP (Ours) | • | 79.60 | 81.36 | • | 65.70 | 74.37 | • | 85.17 | 90.08 | • | 80.67 | 84.22 |

and pseudo-target according to the similarity between the downstream data and each subset. For **domain generalization**, we simply use the first half of the source classes as the pseudo-source and the second half as the pseudo-target.

Within each fold, we lightly fine-tune the baseline model on the pseudo-base/source (training details follow the original paper), then perform a grid search on the pseudo-target to select $\tilde{\tau}$ and $\delta$. We flip predictions according to Equation (11) and the decision rule, aiming to maximize flip coverage while minimizing the erroneous-flip rate. See the appendix for more details.

### 4.2. Experimental Results

**Base-to-Novel Generalization**  As shown in Table 1, we provide detailed results for Base-to-Novel setting across 11 datasets, along with the *balanced harmonic mean (HM)* of their accuracies. Results are encouraging: across four

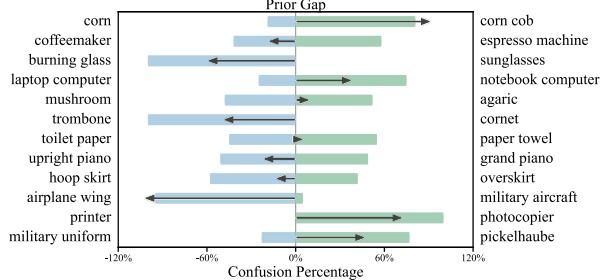

*Figure 5.* Bars to the right of the y-axis show the fraction of samples from the left class misclassified as the right class, and bars to the left show the converse; arrow length denotes the magnitude of the prior gap.

stylistically distinct SOTA baselines, incorporating NeRP consistently improves performance on novel classes. On EuroSAT, several methods gain over 5%; on Flowers102, DTD, and ImageNet, we observe 0.5–2% improvements.

*Table 2.* Cross-dataset generalization experiments on 10 datasets.

| Method | Source | Target Dataset | | | | | | | | | | Average |
| | Image Net | Caltech 101 | Oxford Pets | Stanford Cars | Flowers 102 | Food101 | FGVC Aircraft | SUN397 | DTD | Euro SAT | UCF101 | |
|---|---|---|---|---|---|---|---|---|---|---|---|---|
| CoCoOp | 71.02 | 94.43 | 90.14 | 65.32 | 71.88 | 86.06 | 22.94 | 67.36 | 45.73 | 45.37 | 68.21 | 65.74 |
| +NeRP (Ours) | ● | 94.87 | 90.73 | 65.35 | 72.53 | 86.30 | 23.70 | 67.43 | 46.70 | 51.20 | 68.97 | **66.78** |
| PromptSRC | 71.27 | 93.60 | 90.25 | 65.70 | 70.25 | 86.15 | 23.90 | 67.10 | 46.87 | 45.50 | 68.75 | 65.81 |
| +NeRP (Ours) | ● | 94.10 | 90.30 | 66.20 | 71.37 | 86.40 | 24.23 | 67.30 | 47.83 | 51.70 | 69.70 | **66.91** |
| MMA | 71.00 | 93.80 | 90.30 | 66.13 | 72.07 | 86.12 | 25.33 | 68.17 | 46.57 | 49.24 | 68.32 | 66.61 |
| +NeRP (Ours) | ● | 94.27 | 90.47 | 66.53 | 73.10 | 86.12 | 25.87 | 68.25 | 47.03 | 53.50 | 69.80 | **67.49** |
| MMRL | 72.03 | 94.67 | 91.43 | 66.10 | 72.77 | 86.40 | 26.30 | 67.57 | 45.90 | 53.10 | 68.27 | 67.25 |
| +NeRP (Ours) | ● | 94.77 | 91.43 | 66.60 | 73.97 | 86.50 | 26.50 | 67.73 | 46.17 | 55.87 | 70.43 | **68.00** |

*Table 3.* Domain generalization experiments on 4 datasets.

| Method | Source | Target Dataset | | | | Average |
| | ImageNet | -V2 | -S | -A | -R | |
|---|---|---|---|---|---|---|
| CoCoOp | 71.02 | 64.07 | 48.75 | 50.63 | 76.18 | 59.91 |
| +NeRP (Ours) | ● | 66.73 | 50.20 | 52.47 | 77.00 | **61.60** |
| PromptSRC | 71.27 | 64.35 | 49.55 | 50.90 | 77.80 | 60.65 |
| +NeRP (Ours) | ● | 66.87 | 49.77 | 52.77 | 78.13 | **61.89** |
| MMA | 71.00 | 64.33 | 49.13 | 51.12 | 77.32 | 60.48 |
| +NeRP (Ours) | ● | 67.77 | 50.07 | 51.77 | 78.20 | **61.95** |
| MMRL | 72.03 | 64.47 | 49.17 | 51.12 | 77.32 | 60.52 |
| +NeRP (Ours) | ● | 66.63 | 49.27 | 53.33 | 77.93 | **61.79** |

Because our method adjusts predictions only at inference time, it enhances novel-class accuracy without degrading the performance on base classes.

**Cross-Dataset Evaluation**   Table 2 shows our NeRP method performs well in the case of cross-dataset generalization. Across the four compared methods, adding NeRP yields an average improvement of 1% over 10 datasets, with negligible additional overhead.

**Domain Generalization**   As summarized in Table 3, NeRP stands out in domain generalization settings: on nearly every dataset, plugging in NeRP yields immediate gains. This provides compelling evidence that pairing NeRP with existing methods can serve as a 'free lunch' for efficient transfer learning.

### 4.3. Further Analysis

**Effectiveness of Neutral Priors**   We select the top-ranked misclassified class pairs on ImageNet and evaluate the accuracy of neutral-prior probing under symmetric vs. asymmetric confusions (see Figure 5). The prior direction aligns with the model's bias direction, and its magnitude is small in the symmetric-confusion regime. Across datasets, the top-20% erroneously predicted asymmetric class-confusion pairs (proportion gap >15%) align with the prior direction with 79.7% average consistency.

**Different Calibration/Generation Strategy**   We adopt a unified calibration hyperparameter search as Table 4 shows (10 datasets averaged results). The grid range is set by summarizing model outputs on a small validation subset. Only a single forward pass on $E$ validation splits is needed; af-

*Table 4.* Ablation on search granularity and number of folds $E$. Metrics are accuracy and flip error rate (FER) (%).

| MMRL+NeRP | | | $E=2$ | | $E=3$ | | $E=4$ | | $E=5$ | |
| Step of | $\tilde{\tau}$ | $\delta$ | Acc. | FER | Acc. | FER | Acc. | FER | Acc. | FER |
|---|---|---|---|---|---|---|---|---|---|---|
| Coarse | 1e-2 | 1e-2 | 76.80 | 40.25 | 76.90 | 33.58 | 77.03 | 28.66 | 77.13 | 25.44 |
| Fine | 1e-3 | 1e-3 | 76.88 | 36.15 | 77.04 | 28.02 | 77.16 | 23.87 | **77.32** | **19.91** |

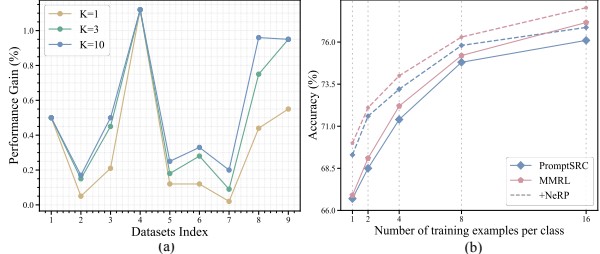

*Figure 6.* (a) Performance of different $K$. (b) Few-shot performance on unseen classes.

ter recording statistics, the search adds negligible overhead. We also study the performance gain of different numbers of confusable pairs across 10 datasets in Figure 6 (a) (EuroSAT excluded due to its few classes). On datasets with high inter-class separability, a smaller $K$ is sufficient; for some datasets, setting $K = 10$ yields larger gains.

**Few-shot Performance**   NeRP remains effective after fine-tuning with fewer samples; results on unseen classes are shown in Figure 6 (b). We still have 16 samples per class, but only a few of them are used for fine-tuning, which serve as a validation set to calibrate $\tilde{\tau}$ and $\delta$.

## 5. Conclusion

We identify an asymmetric confusion phenomenon in VLMs under transfer learning and analyze its downstream impact on BNT. Through theoretical and empirical analyses, we further show that a neutral prior can serve as an anchoring signal that reveals model bias. Building on this insight, we introduce NeRP, a plug-and-play correction strategy to mitigate BNT. Extensive experiments demonstrate that NeRP yields consistent improvements across diverse baselines and datasets.

## Impact Statement

This paper presents work whose goal is to advance the field of Machine Learning. There are many potential societal consequences of our work, none which we feel must be specifically highlighted here.

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

# A. Proof and Supplementary Explanations

## A.1. Turning Zero-shot Stability into a Lemma

**Setup.** Recall the notation from the main paper: for any two novel classes $(i, j) \in \binom{\mathcal{N}}{2}$,

$$\Delta_{ij}^0 := t^0(i) - t^0(j), \qquad \delta_{ij}^0(v) := \langle v, \Delta_{ij}^0 \rangle,$$

and for a domain $D$ we write $\Omega_D := \mathrm{Cov}_{x \sim D}[f_{\mathrm{img}}^0(x)]$. Let $\mathbb{S}^{d-1} := \{v \in \mathbb{R}^d : \|v\|_2 = 1\}$ denote the unit sphere. (Throughout vMF-related statements, we denote the mean direction by $m$ to avoid confusion with $\mu_{ij}(D)$.) Assumption 3.2 asks for a uniform, domain-insensitive upper bound

$$\mathrm{Var}_{x \sim D}\big[\delta_{ij}^0(f_{\mathrm{img}}^0(x))\big] = (\Delta_{ij}^0)^\top \Omega_D \Delta_{ij}^0 \leq \sigma^2,$$

for typical confusable pairs $(i, j) \in \binom{\mathcal{N}}{2}$. We now provide *sufficient conditions* under which the assumption holds as a lemma.

**Lemma A.1** (Uniform projection-variance bound). *Let $Y := f_{\mathrm{img}}^0(x) \in \mathbb{S}^{d-1}$ denote the zero-shot image embedding for $x \sim D$. Suppose one of the following (domain-insensitive) conditions holds:*

(A) ***Lipschitz + sub-Gaussian input.*** *There exists a preprocessing map followed by the zero-shot encoder $F : \mathcal{X} \to \mathbb{S}^{d-1}$ that is $L$-Lipschitz (w.r.t. $\ell_2$), and the preprocessed input distribution has sub-Gaussian parameter $K$ independent of $D$ (i.e., $\langle u, X \rangle$ is $K$-sub-Gaussian for all unit $u$). Then for some absolute constant $C > 0$,*

$$\lambda_{\max}(\Omega_D) \leq C L^2 K^2.$$

(B) ***Spherical concentration (vMF family).*** *There exist $m \in \mathbb{S}^{d-1}$ and $\kappa(D) \geq \kappa_{\min} > 0$ such that $Y$ is distributed according to a von Mises–Fisher family $\mathrm{vMF}(m, \kappa(D))$ (or any spherical concentration family with the same second-moment control). Then there is a dimension/concentration dependent constant $c_d(\kappa_{\min}) < 1$ such that $\lambda_{\max}(\Omega_D) \leq c_d(\kappa_{\min})$, uniformly over $D$.*

(C) ***Restricted eigenvalue with incoherence of inter-class directions.*** *Let $\Omega_D = U \Lambda U^\top$ with eigenvalues $\lambda_1(D) \geq \cdots \geq \lambda_d(D)$. Fix $k$ and assume $\sup_D \lambda_1(D) \leq \Lambda_1$ and $\sup_D \lambda_{k+1}(D) \leq \Lambda_{k+1}$. For each domain $D$, let $T_D$ be the span of the top-$k$ eigenvectors of $\Sigma_D$, and define*

$$\rho := \sup_D \max_{\Delta \in \mathcal{K}} \|\Pi_{T_D} \Delta\|,$$

$$\mathcal{K} := \left\{ \frac{\Delta_{ij}^0}{\|\Delta_{ij}^0\|} : (i, j) \in \binom{\mathcal{N}}{2} \right\}.$$

*If $\rho < 1$, then for all $\Delta \in \mathcal{K}$, $\Delta^\top \Omega_D \Delta \leq \rho^2 \Lambda_1 + (1 - \rho^2) \Lambda_{k+1}$.*

*In each case there exists a domain-insensitive constant $\sigma^2$ such that, for typical $(i, j) \in \binom{\mathcal{N}}{2}$,*

$$\mathrm{Var}_{x \sim D}\big[\delta_{ij}^0(f_{\mathrm{img}}^0(x))\big] = (\Delta_{ij}^0)^\top \Omega_D \Delta_{ij}^0 \leq \sigma^2.$$

*Consequently, Assumption 3.2 holds as a lemma under any of (A)–(C).*

**Proof.** By the Rayleigh-quotient inequality,

$$(\Delta_{ij}^0)^\top \Omega_D \Delta_{ij}^0 \leq \lambda_{\max}(\Omega_D) \|\Delta_{ij}^0\|^2.$$

Since $t^0(i), t^0(j) \in \mathbb{S}^{d-1}$, we have

$$\|\Delta_{ij}^0\| \leq \|t^0(i)\| + \|t^0(j)\| = 2.$$

Thus it suffices to upper bound $\lambda_{\max}(\Omega_D)$ uniformly in $D$ (the factor $\|\Delta_{ij}^0\|^2 \leq 4$ is an absolute constant and can be absorbed into $\sigma^2$).

*(A) Lipschitz + sub-Gaussian.* Let $X$ denote the preprocessed input and $Y = F(X)$. For any unit $u$, the scalar $u^\top Y$ is an $L$-Lipschitz function of $X$. If $X$ is $K$-sub-Gaussian uniformly over $D$, then by standard Lipschitz concentration, $u^\top Y$ is $C_1 L K$-sub-Gaussian for an absolute constant $C_1$. Hence

$$\mathrm{Var}(u^\top Y) \le C_2 L^2 K^2$$

uniformly in $u$ and $D$, implying $\lambda_{\max}(\Omega_D) \le C_2 L^2 K^2$. Set $\sigma^2 = 4 C_2 L^2 K^2$.

*(B) Spherical concentration.* If $Y \sim \mathrm{vMF}(m, \kappa)$ on $\mathbb{S}^{d-1}$ with $\kappa \ge \kappa_{\min} > 0$, the covariance of $Y$ admits a uniform spectral bound depending only on $d$ and $\kappa_{\min}$ (via standard second-moment formulas for vMF). Therefore $\lambda_{\max}(\Omega_D) \le c_d(\kappa_{\min}) < 1$ uniformly over $D$, yielding $\sigma^2 = 4\, c_d(\kappa_{\min})$.

*(C) Restricted eigenvalue + incoherence.* Let $T_D$ be the top-$k$ eigenspace of $\Omega_D$. Decompose any unit $\Delta \in \mathcal{K}$ as

$$\Delta = \Pi_{T_D} \Delta + \Pi_{T_D^\perp} \Delta$$

with $\|\Pi_{T_D} \Delta\| \le \rho$ and $\|\Pi_{T_D^\perp} \Delta\|^2 \ge 1 - \rho^2$. Then

$$\Delta^\top \Omega_D \Delta \le \lambda_1(D) \|\Pi_{T_D} \Delta\|^2 + \lambda_{k+1}(D) \|\Pi_{T_D^\perp} \Delta\|^2$$
$$\le \rho^2 \Lambda_1 + (1 - \rho^2) \Lambda_{k+1}.$$

Taking $\sigma^2 := 4\big(\rho^2 \Lambda_1 + (1 - \rho^2)\Lambda_{k+1}\big)$ completes the argument. $\qquad\square$

**Corollary A.2** (Low-rank domain shift). $f\, \Omega_D = \Omega_0 + E_D$ with $\|E_D\|_{\mathrm{op}} \le \varepsilon$ *uniformly in D, then*

$$\lambda_{\max}(\Omega_D) \ \le\ \lambda_{\max}(\Omega_0) + \varepsilon.$$

*In particular, any bound $\lambda_{\max}(\Omega_0) \le \bar\sigma^2$ lifts to $\lambda_{\max}(Omega_D) \le \bar\sigma^2 + \varepsilon$, uniformly in D.*

**Proof.** By Weyl's inequality for symmetric matrices,

$$\lambda_{\max}(\Omega_0 + E_D) \le \lambda_{\max}(\Omega_0) + \lambda_{\max}(E_D)$$
$$\le \lambda_{\max}(\Omega_0) + \|E_D\|_{\mathrm{op}}$$
$$\le \lambda_{\max}(\Omega_0) + \varepsilon.$$

$$\square$$

**Corollary A.3** (Bound with residualization). *Let $P := I - \Pi_W$ be the orthogonal projector that removes a (domain) anchor subspace $W \subset \mathbb{R}^d$. Define the residualized feature $\widetilde{Y} := PY$ and covariance $\widetilde{\Omega}_D := P\Omega_D P$. Then for all domains $D$, $\lambda_{\max}(\widetilde{\Omega}_D) \ \le\ \lambda_{\max}(\Omega_D)$. Hence replacing $\delta_{ij}^0(Y)$ by $\delta_{ij}^0(\widetilde{Y})$ weakens or preserves the variance bound $\sigma^2$.*

**Proof.** For any unit $u$, $u^\top \widetilde{\Omega}_D u = (Pu)^\top \Omega_D (Pu) \le \lambda_{\max}(\Omega_D) \|Pu\|^2 \le \lambda_{\max}(\Omega_D)$. Taking the supremum over $u$ yields $\lambda_{\max}(\widetilde{\Omega}_D) \le \lambda_{\max}(\Omega_D)$. $\qquad\square$

### A.2. Proof sketch of Lemma 3.3.

*Image prior.* By definition,
$$\Delta\pi_{\mathrm{img}}(i,j) = \big\langle u_{\mathrm{img}}(D),\, t^0(i)\big\rangle - \big\langle u_{\mathrm{img}}(D),\, t^0(j)\big\rangle$$
$$= \big\langle u_{\mathrm{img}}(D),\, \Delta_{ij}^0\big\rangle,$$

which proves the first identity.

*Text prior.* Let $g_c^0 := g_{\mathrm{txt}}^0(\mathrm{prompt}_0(c))$ so that $t^0(c) = \mathrm{norm}(g_c^0)$, and recall the low-rank raw update $g_{\mathrm{txt}}(\mathrm{prompt}(c)) = g_c^0 + U_{\mathrm{txt}}\, b(c)$. Write $h(v) := v/\|v\|$ so $t(c) = h\big(g_c^0 + U_{\mathrm{txt}} b(c)\big)$. A first-order expansion of $h$ at $g_c^0$ yields

$$t(c) = t^0(c) + J_{g_c^0}\big(U_{\mathrm{txt}} b(c)\big) + r_c,$$
$$J_{g_c^0} = \frac{1}{\|g_c^0\|}\Big(I - t^0(c)\, t^0(c)^\top\Big),$$

with a quadratic remainder satisfying $\|r_c\| \lesssim \|U_{\text{txt}}b(c)\|^2/\|g_c^0\|^2$. Hence

$$
\begin{aligned}
\Delta\pi_{\text{txt}}(i,j) &= \langle t(i) - t(j),\, u_{\text{txt}}^0(D)\rangle \\
&= \langle t^0(i) - t^0(j),\, u_{\text{txt}}^0(D)\rangle \\
&\quad + \langle J_{g_i^0}U_{\text{txt}}b(i) - J_{g_j^0}U_{\text{txt}}b(j),\, u_{\text{txt}}^0(D)\rangle \\
&\quad + \langle r_i - r_j,\, u_{\text{txt}}^0(D)\rangle \\
&= \underbrace{\langle \Delta_{ij}^0,\, u_{\text{txt}}^0(D)\rangle}_{\text{pretrained term}} \\
&\quad + \underbrace{\langle U_{\text{txt}}\big(b(i) - b(j)\big),\, u_{\text{txt}}^0(D)\rangle}_{\text{dominant correction}} \\
&\quad + \varepsilon_{ij}.
\end{aligned}
$$

where we used that $J_{g_c^0}$ is the identity on the tangent space at $t^0(c)$ (and $u_{\text{txt}}^0(D)$ is unit-norm), and absorbed the deviation of $J_{g_c^0}$ from the identity and the quadratic remainders into $\varepsilon_{ij}$. Using $\|J_{g_c^0}\| \leq 1/\|g_c^0\|$, $\|u_{\text{txt}}^0(D)\| = 1$, and the bound on $\|r_c\|$, we obtain, under the standard pretrained-norm condition $\inf_c \|g_c^0\| \geq c_0 > 0$,

$$
\begin{aligned}
|\varepsilon_{ij}| &\lesssim \frac{\|U_{\text{txt}}b(i)\|}{\|g_i^0\|} + \frac{\|U_{\text{txt}}b(j)\|}{\|g_j^0\|} \\
&\quad + \frac{\|U_{\text{txt}}b(i)\|^2}{\|g_i^0\|^2} + \frac{\|U_{\text{txt}}b(j)\|^2}{\|g_j^0\|^2} \\
&\lesssim \|U_{\text{txt}}\|\big(\|b(i)\| + \|b(j)\|\big),
\end{aligned}
$$

which gives the stated deviation bound

$$
\begin{aligned}
&\big|\Delta\pi_{\text{txt}}(i,j) - \langle \Delta_{ij}^0,\, u_{\text{txt}}^0(D)\rangle\big| \\
&\lesssim \|U_{\text{txt}}\|\big(\|b(i)\| + \|b(j)\|\big).
\end{aligned}
$$

Therefore the image prior gap is exactly the projection of $u_{\text{img}}(D)$ onto $\Delta_{ij}^0$, and the text prior gap coincides with the projection of $\Delta_{ij}^0$ onto $u_{\text{txt}}^0(D)$ up to a controlled novel-class residual driven by the low-rank text update. $\qquad\square$

### A.3. Proof of Lemma 3.4.

Assume the anchor projections satisfy $\|\Pi_S u_{\text{img}}(D)\| \leq \kappa_{\text{img}}$ and $\|\Pi_S u_{\text{txt}}^0(D)\| \leq \kappa_{\text{txt}}$. Recall $S = \text{span}\{t^0(c) : c \in \mathcal{B}\}$ and that $\Delta_{ij}^0 = t^0(i) - t^0(j) \in S$ for $i, j \in \mathcal{B}$.

*Image side.* By definition and orthogonal decomposition,

$$
\Delta\pi_{\text{img}}(i,j) = \big\langle u_{\text{img}}(D),\, \Delta_{ij}^0\big\rangle = \big\langle \Pi_S u_{\text{img}}(D),\, \Delta_{ij}^0\big\rangle,
$$

since $\Pi_{S^\perp}\Delta_{ij}^0 = 0$. By Cauchy–Schwarz,

$$
\begin{aligned}
|\Delta\pi_{\text{img}}(i,j)| &\leq \|\Pi_S u_{\text{img}}(D)\|\,\|\Delta_{ij}^0\| \\
&= \|\Pi_S u_{\text{img}}(D)\|\,\|\Pi_S \Delta_{ij}^0\| \\
&\leq \kappa_{\text{img}}\|\Pi_S \Delta_{ij}^0\|.
\end{aligned}
$$

*Text side.* Decompose both factors into $S \oplus S^\perp$:

$$
\begin{aligned}
\Delta\pi_{\text{txt}}(i,j) &= \big\langle t(i) - t(j),\, u_{\text{txt}}^0(D)\big\rangle \\
&= \big\langle \Pi_S\big(t(i) - t(j)\big),\, \Pi_S u_{\text{txt}}^0(D)\big\rangle \\
&\quad + \big\langle \Pi_{S^\perp}\big(t(i) - t(j)\big),\, \Pi_{S^\perp} u_{\text{txt}}^0(D)\big\rangle.
\end{aligned}
$$

Applying Cauchy–Schwarz to each term and the anchor bound on $S$,

$$
\begin{aligned}
|\Delta\pi_{\text{txt}}(i,j)| &\leq \|\Pi_S u^0_{\text{txt}}(D)\| \, \|\Pi_S\big(t(i)-t(j)\big)\| \\
&\quad + \|\Pi_{S^\perp} u^0_{\text{txt}}(D)\| \, \|\Pi_{S^\perp}\big(t(i)-t(j)\big)\| \\
&\leq \kappa_{\text{txt}} \, \|\Pi_S(t(i)-t(j))\| \\
&\quad + \|\Pi_{S^\perp} u^0_{\text{txt}}(D)\| \, \|\Pi_{S^\perp}(t(i)-t(j))\|.
\end{aligned}
$$

Combining the two parts yields the claimed bounds. $\qquad\square$

### A.4. Proof of Proposition 3.5.

Fix a novel pair $(i,j) \in \mathcal{N} \times \mathcal{N}$. Recall

$$
\mu_{ij}(D) = \mathbb{E}_{x\sim D}\big[\langle f_{\text{img}}(x),\, t(i)-t(j)\rangle\big]
$$

and the gaps

$$
\Delta\pi_{\text{img}}(i,j) = \langle u_{\text{img}}(D),\, \Delta^0_{ij}\rangle,
$$
$$
\begin{aligned}
\Delta\pi_{\text{txt}}(i,j) =& \langle \Delta^0_{ij},\, u^0_{\text{txt}}(D)\rangle \\
&+ \langle U_{\text{txt}}(b(i)-b(j)),\, u^0_{\text{txt}}(D)\rangle
\end{aligned}
$$

from Lemma 3.3.

*Step 1: Linearization around pretrained geometry.* By the low-rank deformation model and smoothness of normalization on the sphere, write

$$
f_{\text{img}}(x) = f^0_{\text{img}}(x) + \delta_{\text{img}}(x),
$$
$$
t(i)-t(j) = \Delta^0_{ij} + \delta_{\text{txt}}(i,j),
$$

where $\mathbb{E}\|\delta_{\text{img}}(x)\|$ is small on $\mathcal{N}$ and $\|\delta_{\text{txt}}(i,j)\|$ is small since $b(i), b(j)$ are small on $\mathcal{N}$. Then

$$
\begin{aligned}
\mu_{ij}(D) =& \mathbb{E}\langle f^0_{\text{img}}(x), \Delta^0_{ij}\rangle + \mathbb{E}\langle f^0_{\text{img}}(x), \delta_{\text{txt}}(i,j)\rangle \\
&+ \mathbb{E}\langle \delta_{\text{img}}(x), \Delta^0_{ij}\rangle + \mathbb{E}\langle \delta_{\text{img}}(x), \delta_{\text{txt}}(i,j)\rangle.
\end{aligned}
$$

By Cauchy–Schwarz and the smallness of $\delta_{\text{img}}, \delta_{\text{txt}}$, there exists $\varepsilon^{\mathcal{N}}_{\text{deform}} > 0$ (small on $\mathcal{N}$) such that

$$
\big|\mu_{ij}(D) - \mathbb{E}\langle f^0_{\text{img}}(x), \Delta^0_{ij}\rangle\big| \;\leq\; \varepsilon^{\mathcal{N}}_{\text{deform}}. \tag{13}
$$

*Step 2: Expected zero-shot margin vs. image prior gap.* Let $\bar{x}^D := \mathbb{E}_{x\sim D}[x]$ and $\Omega_x := \text{Cov}_{x\sim D}[x]$. Define

$$
h^0(x) := f^0_{\text{img}}(x) = \text{norm}\big(g^0_{\text{img}}(x)\big),
$$
$$
u_{\text{img}}(D) := f_{\text{img}}(\bar{x}^D).
$$

Assume $h^0$ is twice continuously differentiable on $\mathcal{K} := \text{conv}(\text{supp}(D))$, and there exists a curvature bound

$$
M^0_{\text{img}} := \sup_{\substack{z\in\mathcal{K} \\ \|v\|_2=1}} \big\|\nabla^2\langle h^0(z),\, v\rangle\big\|_{\text{op}} < \infty. \tag{14}
$$

For notational simplicity we take $\|\Delta^0_{ij}\|_2 = 1$ (otherwise multiply the right-hand sides below by $\|\Delta^0_{ij}\|_2$). By the second-order Taylor expansion with mean-value remainder applied to the scalar field $\phi(x) := \langle h^0(x), \Delta^0_{ij}\rangle$, we have

$$
\big| \mathbb{E}\langle h^0(x), \Delta^0_{ij}\rangle - \langle h^0(\bar{x}^D), \Delta^0_{ij}\rangle \big| \;\leq\; \frac{1}{2} M^0_{\text{img}} \text{Tr}(\Omega_x). \tag{15}
$$

Moreover, at the anchor point $\bar{x}^D$ the (fine-tuned vs. zero-shot) pointwise deformation contributes

$$
\begin{aligned}
\big| \langle h^0(\bar{x}^D) &- f_{\text{img}}(\bar{x}^D),\, \Delta^0_{ij}\rangle \big| \\
&\leq \big\| f_{\text{img}}(\bar{x}^D) - h^0(\bar{x}^D)\big\|_2 \\
&=: \varepsilon^{\mathcal{N}}_{\text{pt}}.
\end{aligned} \tag{16}
$$

Combining (15) and (16), we obtain the explicit curvature-controlled anchor bound

$$\left| \mathbb{E}\langle f^0_{\mathrm{img}}(x), \Delta^0_{ij}\rangle - \langle u_{\mathrm{img}}(D), \Delta^0_{ij}\rangle \right|$$
$$\leq \frac{1}{2} M^0_{\mathrm{img}} \operatorname{Tr}(\Omega_x) + \varepsilon^{\mathcal{N}}_{\mathrm{pt}}. \tag{17}$$

We may therefore set

$$\varepsilon^{\mathcal{N}}_{\mathrm{img,anch}} := \tfrac{1}{2} M^0_{\mathrm{img}} \operatorname{Tr}(\Omega_x) + \varepsilon^{\mathcal{N}}_{\mathrm{pt}}.$$

*Step 3: Expected zero-shot margin vs. text prior gap.* By Lemma 3.3,

$$\Delta\pi_{\mathrm{txt}}(i,j) = \langle \Delta^0_{ij}, u^0_{\mathrm{txt}}(D)\rangle$$
$$+ \langle U_{\mathrm{txt}}\big(b(i) - b(j)\big), u^0_{\mathrm{txt}}(D)\rangle.$$

The cross-modal alignment between the zero-shot image mean and the neutral text anchor along $\Delta^0_{ij}$ yields a small constant $\varepsilon^{\mathcal{N}}_{\mathrm{txt,align}} > 0$ such that

$$\left| \mathbb{E}\langle f^0_{\mathrm{img}}(x), \Delta^0_{ij}\rangle - \langle \Delta^0_{ij}, u^0_{\mathrm{txt}}(D)\rangle \right| \leq \varepsilon^{\mathcal{N}}_{\mathrm{txt,align}}. \tag{18}$$

Moreover, the low-rank text residual satisfies $\left|\langle U_{\mathrm{txt}}(b(i) - b(j)), u^0_{\mathrm{txt}}(D)\rangle\right| \leq \|U_{\mathrm{txt}}\|(\|b(i)\| + \|b(j)\|)$, which is small on $\mathcal{N}$. Using (13) and (18) again by triangle inequality,

$$\left|\mu_{ij}(D) - \Delta\pi_{\mathrm{txt}}(i,j)\right|$$
$$\leq \underbrace{\varepsilon^{\mathcal{N}}_{\mathrm{deform}} + \varepsilon^{\mathcal{N}}_{\mathrm{txt,align}} + \|U_{\mathrm{txt}}\|(\|b(i)\| + \|b(j)\|)}_{:= \varepsilon^{\mathcal{N}}_{\mathrm{txt}}}.$$

*Step 4: Sign consistency.* By the two bounds above, whenever $|\Delta\pi_{\mathrm{img}}(i,j)| > \varepsilon^{\mathcal{N}}_{\mathrm{img}}$ (respectively $|\Delta\pi_{\mathrm{txt}}(i,j)| > \varepsilon^{\mathcal{N}}_{\mathrm{txt}}$), the scalar $\mu_{ij}(D)$ has the same sign as $\Delta\pi_{\mathrm{img}}(i,j)$ (resp. $\Delta\pi_{\mathrm{txt}}(i,j)$), since perturbations smaller than the magnitude cannot flip sign. This proves the proposition with

$$\varepsilon^{\mathcal{N}}_{\mathrm{img}} = \varepsilon^{\mathcal{N}}_{\mathrm{deform}} + \varepsilon^{\mathcal{N}}_{\mathrm{img,anch}}$$

and

$$\varepsilon^{\mathcal{N}}_{\mathrm{txt}} = \varepsilon^{\mathcal{N}}_{\mathrm{deform}} + \varepsilon^{\mathcal{N}}_{\mathrm{txt,align}} + \|U_{\mathrm{txt}}\|(\|b(i)\| + \|b(j)\|).$$

$\square$

## A.5. Proof of Corollary 3.6.

Fix a novel pair $(i,j) \in \mathcal{N} \times \mathcal{N}$. Let

$$m := m_{ij}(x) = \langle f_{\mathrm{img}}(x), t(i) - t(j)\rangle,$$

$$\mu := \mu_{ij}(D) = \mathbb{E}[m],$$

and $\sigma^2_m := \operatorname{Var}[m]$. Under Assumption 3.2 and the discussion following Proposition 3.5, we have $\sigma^2_m \lesssim \sigma^2$ on $\mathcal{N}$.

*Step 1: Sign error implies a large deviation.* Assume w.l.o.g. $\mu > 0$ (the case $\mu < 0$ is symmetric). If $\operatorname{sign} m \neq \operatorname{sign} \mu$ then $m \leq 0$, hence $|m - \mu| = \mu - m \geq \mu = |\mu|$. For any $\gamma \in (0, |\mu|)$,

$$\{\operatorname{sign} m \neq \operatorname{sign} \mu\} \subseteq \{|m - \mu| \geq |\mu|\}$$
$$\subseteq \{|m - \mu| \geq |\mu| - \gamma\}.$$

*Step 2: Chebyshev's inequality.* By Chebyshev,

$$\mathbb{P}\big(|m - \mu| \geq |\mu| - \gamma\big) \leq \frac{\sigma^2_m}{(|\mu| - \gamma)^2}.$$

Combining with Step 1 gives

$$\mathbb{P}\big[\operatorname{sign} m \neq \operatorname{sign} \mu\big] \leq \frac{\sigma_m^2}{(\,|\mu| - \gamma\,)^2}$$
$$\leq \min\left\{\frac{\sigma_m^2}{(\,|\mu| - \gamma)^2}, 1\right\}, \tag{19}$$

whenever $|\mu| > \gamma$.

*Step 3: Substitute prior gaps via Proposition 3.5.* On $\mathcal{N}$, Proposition 3.5 yields

$$|\mu - \Delta\pi_{\mathrm{img}}(i,j)| \leq \varepsilon_{\mathrm{img}}^{\mathcal{N}}$$

and

$$|\mu - \Delta\pi_{\mathrm{txt}}(i,j)| \leq \varepsilon_{\mathrm{txt}}^{\mathcal{N}}.$$

Thus, if $|\Delta\pi_{\mathrm{img}}(i,j)| > \varepsilon_{\mathrm{img}}^{\mathcal{N}} + \gamma$ (respectively, if $|\Delta\pi_{\mathrm{txt}}(i,j)| > \varepsilon_{\mathrm{txt}}^{\mathcal{N}} + \gamma$), then $|\mu| \geq |\Delta\pi| - \varepsilon > \gamma$ and (19) implies

$$\mathbb{P}\big[\operatorname{sign} m \neq \operatorname{sign} \mu\big] \leq \frac{\sigma_m^2}{\big(\,|\Delta\pi| - \varepsilon - \gamma\,\big)^2}.$$

Equivalently,

$$\mathbb{P}\big[\operatorname{sign} m = \operatorname{sign} \Delta\pi\big] \geq 1 - \frac{\sigma_m^2}{\big(\,|\Delta\pi| - \varepsilon - \gamma\,\big)^2},$$

with $\varepsilon = \varepsilon_{\mathrm{img}}^{\mathcal{N}}$ for the image prior (resp. $\varepsilon_{\mathrm{txt}}^{\mathcal{N}}$ for the text prior). This establishes the claimed high-probability sign consistency.
$\square$

## A.6. Closed-form of Intercept.

Recall Eq. (5):

$$\hat{\beta}(D) := \underset{\beta}{\operatorname{argmin}} \sum_{(i,j)\in\mathcal{B}\times\mathcal{B}} \big(\hat{\mu}_{ij}(D) - \Sigma_{i,j}(D) - \beta\big)^2.$$

We can fit the intercept over a generated base-pair set

$$E_B \subseteq (i,j) \in \mathcal{B} \times \mathcal{B} : i < j,$$

so that each unordered pair appears exactly once. Let $M := |E_B|$. Define $t_{ij} := \hat{\mu}_{ij}(D)$ and $s_{ij} := \Sigma_{i,j}(D)$. The one-dimensional least-squares objective

$$J(\beta) = \sum_{(i,j)\in E_B} (t_{ij} - s_{ij} - \beta)^2$$

is strictly convex, and the first-order optimality condition $\partial_\beta J(\beta) = 0$ yields the closed form

$$\boxed{\beta^\star(D) = \frac{1}{M} \sum_{(i,j)\in E_B} \big(\hat{\mu}_{ij}(D) - \Sigma_{i,j}(D)\big)}.$$

Intuitively, $\beta^\star(D)$ is the average residual (data mean gap minus composite prior gap) across oriented pairs.

*Empirical estimation of $\hat{\mu}_{ij}(D)$.* Let $X_D$ be a calibration split from domain $D$ (e.g., held-out or training images after preprocessing). For each $(i,j) \in E_B$,

$$\hat{\mu}_{ij}(D) := \frac{1}{|X_D|} \sum_{x \in X_D} m_{ij}(x),$$
$$m_{ij}(x) = \langle f_{\mathrm{img}}(x),\, t(i) - t(j) \rangle.$$

This is label-free (it uses only embeddings and class prototypes), though one may restrict $X_D$ to base-class images if desired.

## A.7. Residual composite prior preserves guarantees.

Recall the residual composite prior in (8):

$$\tilde{\Sigma}_{i,j}(D) = \big(\tilde{\pi}_{\text{txt}}(i; D) - \tilde{\pi}_{\text{txt}}(j; D)\big)$$
$$+ \big(\tilde{\pi}_{\text{img}}(i; D) - \tilde{\pi}_{\text{img}}(j; D)\big),$$

with

$$\tilde{\pi}_{\text{txt}}(c; D) = \langle t(c), u^0_{\text{txt}}(D)\rangle - \langle t(c), u_{\text{txt}}(D)\rangle$$

and

$$\tilde{\pi}_{\text{img}}(c; D) = \langle u_{\text{img}}(D), t^0(c)\rangle - \langle u_{\text{img}}(D), t(c)\rangle.$$

**Proposition A.4** (Residual sign consistency)**.** *Under the low-rank deformation model* (3) *and Assumption 3.2, there exist domain- and model-dependent constants $\tilde{\varepsilon}^{\mathcal{N}}_{\text{img}}$, $\tilde{\varepsilon}^{\mathcal{N}}_{\text{txt}}$ (small on $\mathcal{N}$) such that for any $(i, j) \in \mathcal{N} \times \mathcal{N}$,*

$$\big|\mu_{ij}(D) - \tilde{\Sigma}_{i,j}(D)\big| \leq \tilde{\varepsilon}^{\mathcal{N}} := \tilde{\varepsilon}^{\mathcal{N}}_{\text{img}} + \tilde{\varepsilon}^{\mathcal{N}}_{\text{txt}}.$$

*Consequently,* $\text{sign}\,\mu_{ij}(D) = \text{sign}\,\tilde{\Sigma}_{i,j}(D)$ *whenever* $|\tilde{\Sigma}_{i,j}(D)| > \tilde{\varepsilon}^{\mathcal{N}}$*. Moreover, typically* $\tilde{\varepsilon}^{\mathcal{N}} \leq \varepsilon^{\mathcal{N}}$ *from Proposition 3.5, i.e., the constants are (weakly) tighter.*

**Proof sketch.** Let $d_{\text{txt}} := u^0_{\text{txt}}(D) - u_{\text{txt}}(D)$ and write the fine-tuned inter-class difference as $t(i) - t(j) = \Delta^0_{ij} + \eta_{ij}$ with $\|\eta_{ij}\|$ small on $\mathcal{N}$ (low-rank deformation off the base support). Using Lemma 3.3 and the residual definitions,

$$\Delta\tilde{\pi}_{\text{txt}}(i, j) = \underbrace{\langle\Delta^0_{ij}, d_{\text{txt}}\rangle}_{\text{anchor displacement}}$$
$$+ \underbrace{\langle U_{\text{txt}}(b(i) - b(j)), u^0_{\text{txt}}\rangle}_{\text{novel small}}$$
$$- \underbrace{\langle\eta_{ij}, u_{\text{txt}}\rangle}_{\text{deformation small}},$$
$$\Delta\tilde{\pi}_{\text{img}}(i, j) = \Delta\pi_{\text{img}}(i, j) - \langle u_{\text{img}}, t(i) - t(j)\rangle$$
$$- \langle u_{\text{img}}, \eta_{ij}\rangle.$$

Summing gives

$$\tilde{\Sigma}_{i,j}(D) = \langle\Delta^0_{ij}, d_{\text{txt}}\rangle + \langle U_{\text{txt}}(b(i) - b(j)), u^0_{\text{txt}}\rangle$$
$$- \langle\eta_{ij}, u_{\text{txt}} + u_{\text{img}}\rangle.$$

Since $\|u_{\text{txt}}\| = \|u_{\text{img}}\| = 1$, Cauchy–Schwarz yields

$$|\langle\eta_{ij}, u_{\text{txt}} + u_{\text{img}}\rangle| \leq \sqrt{2}\,\|\eta_{ij}\|.$$

Relating $\tilde{\Sigma}_{i,j}$ to $\mu_{ij}$ via the bound $|\mu_{ij} - \Sigma_{i,j}| \leq \varepsilon^{\mathcal{N}}$ from Proposition 3.5 and the identity

$$\tilde{\Sigma}_{i,j} = \Sigma_{i,j} - \langle t(i) - t(j), u_{\text{txt}}\rangle - \langle u_{\text{img}}, t(i) - t(j)\rangle,$$

we obtain

$$|\mu_{ij} - \tilde{\Sigma}_{i,j}| \leq \varepsilon^{\mathcal{N}} + \sqrt{2}\,\|\eta_{ij}\|$$
$$+ \|U_{\text{txt}}\|\big(\|b(i)\| + \|b(j)\|\big)$$
$$:= \tilde{\varepsilon}^{\mathcal{N}}.$$

On $\mathcal{N}$, both $\|\eta_{ij}\|$ and $\|b(\cdot)\|$ are small, hence $\tilde{\varepsilon}^{\mathcal{N}} \leq \varepsilon^{\mathcal{N}}$ typically. The sign statement follows as in Proposition 3.5.

*Table 5.* Prompt template for the generation of confusable pairs.

---

**System Prompt:**
You are a senior dataset curator for visual classification.

GOAL
* For a given anchor class in a CLOSED label set, propose up to K confusable candidate classes (pairs of [anchor, candidate]) strictly from the provided class list.

CONSTRAINTS
* Only choose from the provided label list. Never invent labels.

CONFUSABILITY CRITERIA (non-exhaustive)
* Visual similarity (shape / texture / color / pose)
* Fine-grained taxonomy proximity (hypernym / hyponym / sibling)
* Name-level ambiguity (homonym / near-spelling)
* Typical co-occurring background / scene
* Similar granularity within this dataset

OUTPUT REQUIREMENTS
* Return JSON ONLY, no prose, no explanations.
* Use EXACT label names as provided (case-sensitive).
* If fewer than K plausible candidates exist, return fewer.
* No duplicates; never include the anchor itself.
* Sort candidates by estimated confusion likelihood (descending).

JSON schema:
```
{
"dataset": "<string>", "anchor": "<string>", "k": <int>,
"candidates": [
  {
    "label": "<one of provided labels>",
    "rationale": "<<= 20 words, concise reason>",
    "mode": "<one of: visual|semantic|name|context>",
    "score": <0–100 integer likelihood>
  }
]
}
```

---

**User Prompt**
Dataset: {DATASET_NAME} ANCHOR: "{ANCHOR_CLASS}" K: {K}

TASK
* From the CLOSED label set below, propose up to K confusable candidate classes for [{ANCHOR_CLASS}], using only labels in the list.
* Prefer candidates that would be genuinely hard for a CLIP-like classifier on {DATASET_NAME}.
* Be conservative and precise.
* Output JSON ONLY per the schema.

LABEL SET (with short descriptions) {LABEL_JSON_ARRAY}
# Example entry format inside LABEL_JSON_ARRAY:
# [
# {"label": "Ketch", "desc": "A type of two-masted sailboat"},
# {"label": "Schooner", "desc": "A sailing ship with two or more masts."}
# ]

---

**Corollary A.5** (Residual high-probability consistency). *Under Assumption 3.2, for $(i,j) \in \mathcal{N} \times \mathcal{N}$ let $\sigma_m^2 := \mathrm{Var}_{x \sim D}[m_{ij}(x)] \lesssim \sigma^2$. Then for any $\gamma > 0$,*

$$\mathbb{P}_{x \sim D}\Big[ \mathrm{sign}\, m_{ij}(x) \neq \mathrm{sign}\, \mu_{ij}(D) \Big]$$

$$\leq \min\left\{ \frac{\sigma_m^2}{\left( |\tilde{\Sigma}_{i,j}(D)| - \tilde{\varepsilon}^{\mathcal{N}} - \gamma \right)^2}, 1 \right\},$$

whenever $|\tilde{\Sigma}_{i,j}(D)| > \tilde{\varepsilon}^{\mathcal{N}} + \gamma$.

**Proof sketch.** The probabilistic step in Corollary 3.6 depends only on the variance of $m_{ij}(x)$, which is unchanged by the choice of prior. Replace $\varepsilon^{\mathcal{N}}$ by $\tilde{\varepsilon}^{\mathcal{N}}$ from Proposition A.4 to obtain the stated bound.

**Why the constants tighten (variance reduction).** Across class pairs $(i, j)$, residualization subtracts the class-agnostic projections onto the *current* anchors:

$$\tilde{\Sigma}_{i,j} = \Sigma_{i,j} - \langle t(i) - t(j), u_{\text{txt}} \rangle - \langle u_{\text{img}}, t(i) - t(j) \rangle.$$

For a random pairwise direction $Z_{ij}$,

$$\text{Var}\big(\langle Z_{ij}, a \rangle - \langle Z_{ij}, b \rangle\big) = \text{Var}\big(\langle Z_{ij}, a - b \rangle\big)$$
$$\leq \text{Var}\big(\langle Z_{ij}, a \rangle\big).$$

Thus, removing the class-independent anchor components reduces the across-pair dispersion of prior gaps while retaining the informative anchor displacement $\langle \Delta_{ij}^{0}, d_{\text{txt}} \rangle$ and only small deformation terms. Empirically, this yields $\tilde{\varepsilon}^{\mathcal{N}}$ that is no larger—and usually smaller—than $\varepsilon^{\mathcal{N}}$, sharpening both the deterministic and high-probability guarantees.

## B. Implementation Details

### B.1. Confusable Pairs Generation

Given the closed label set $\mathcal{C}$ of a downstream task, we pre-compute an *anchor–candidate* confusion graph $G$ that drives all NeRP corrections. The procedure is illustrated in Alg. 1 and summarised below.

For every anchor class $i \in \mathcal{C}$ we query a local LLM `Qwen2.5-72B-Instruct` with the system/user prompt template shown in Table 5. The model is asked to return up to $K$ *confusable* labels from the same set, ranked by predicted confusion likelihood and accompanied by a short rationale. Following practical studies, we set different $K$ for different datasets; larger values bring marginal gains but increase a little bit of latency.

All $\langle i, j \rangle$ pairs emitted by the LLM are aggregated into an undirected graph $G \subseteq \mathcal{C} \times \mathcal{C}$: an edge $(i, j)$ is present iff either direction is proposed and $i \neq j$. Duplicate proposals are ignored, and self-loops are forbidden by construction.

*Table 6.* ImageNet-1K Superclasses and typical examples

| ID | Superclass | Scope (short) | Typical ImageNet-1K examples |
|----|-----------|---------------|------------------------------|
| 1 | Mammals (incl. pets) | Domestic and wild mammals | tabby, Siamese cat, Persian cat |
| 2 | Birds | Avian species (flighted and flightless) | hen, ostrich, bald eagle, king penguin, peacock |
| 3 | Reptiles / Amphibians | Non-avian, cold-blooded vertebrates | alligator, chameleon, Komodo dragon |
| 4 | Fish & Marine life | Fish and marine invertebrates | goldfish, great white shark, stingray |
| 5 | Insects & Invertebrates | Terrestrial arthropods and similar | monarch, ladybug, dragonfly |
| 6 | Plants–Flowers | Flowering plants | daisy, sunflower, lotus, water lily, poppy |
| 7 | Plants–Other | Trees, fruits, crops, fungi | mushroom, banana, pineapple |
| 8 | Food & Beverages | Prepared foods and common edibles | pizza, hot dog, hamburger, ice cream, bagel |
| 9 | Vehicles–Land | Road and land vehicles (incl. 2-wheel) | sports car, jeep, pickup, minivan |
| 10 | Vehicles–Air | Aircraft and lighter-than-air | airliner, warplane, biplane, airship |
| 11 | Vehicles–Water | Watercraft and boats | schooner, speedboat, canoe, lifeboat |
| 12 | Electronics & Imaging | Consumer electronics, imaging devices | laptop, desktop computer, monitor |
| 13 | Household Objects | Furniture, containers, kitchenware | coffee mug, cup, bottle, plate, sofa |
| 14 | Tools & Machinery | Hand tools and powered machines | chain saw, lawn mower, sewing machine, shovel |
| 15 | Sports & Recreation Gear | Sports balls, equipment, protective gear | soccer ball, tennis ball, baseball |
| 16 | Musical Instruments & Audio | Instruments and audio gear | acoustic guitar, violin, grand piano, drum |
| 17 | Buildings & Large Structures | Man-made large structures | castle, church, mosque, lighthouse |
| 18 | Natural Scenes & Landforms | Outdoor scenes and geologic features | seashore, lakeside, valley, cliff |
| 19 | Apparel & Personal Gear | Clothing, footwear, accessories | sunglasses, backpack, running shoe, cowboy hat |
| 20 | Materials & Surfaces | Texture-bearing surface objects | tile roof, stone wall, doormat, quilt |

### B.2. Target-Emulated Calibration

Recall that NeRP's decisions rely on two gates: the Prior Effective Gate $\tilde{\tau}$ and the Evidence Strength Gate $\delta$. Because the test data are unseen, we construct simulated subsets from the training data to select appropriate $\tilde{\tau}$ and $\delta$ for each task.

**Algorithm 1** Pseudocode of building graph $G$.

```
def build_graph():
    labels_json = json.dumps(
                ["label": c["label"],
                 "desc": c.get("desc","")
                 for c in ALL_CLASSES],
                ensure_ascii=False)
    edges, per_anchor = set(),
    for cls in ALL_CLASSES:
        anchor = cls["label"]
        user_prompt = render_user_prompt(
                    ds_name,
                    anchor,
                    K,
                    labels_json)
        res = call_qwen(
            system_prompt,
            user_prompt
            )
        per_anchor[anchor] = res
        for cand in res.get("candidates", []):
            l = cand.get("label")
            if l and l != anchor:
                edges.add(tuple(
                        sorted((anchor, l))
                        ))
    return edges, per_anchor
```

Here, we provide the division and attribution of superclasses on the source dataset ImageNet and 10 target datasets. Table 6 consolidates ImageNet-1K into 20 visually and semantically coherent superclasses with typical examples, while Table 7 aligns ten downstream datasets to these buckets using primary matches and nearest-style proxies when 1K lacks a direct label; this organization unifies semantics across fine-grained sets (e.g., cars, aircraft) at a robust coarse level, enables appearance-based alignment for datasets without direct counterparts (textures, aerial tiles, human actions) via buckets such as *Materials & Surfaces*, *Natural Scenes & Landforms*, and *Buildings & Large Structures*, and provides foldable units for cross-dataset simulations. In the base-to-novel setting, we split base classes into $E$ disjoint folds $\{F_e^{\mathrm{b2n}}\}_{e=1}^E$ and choose the Prior Effective Gate $\tilde{\tau}$ and Evidence Strength Gate $\delta$ on simulated subsets; superclass awareness helps maintain semantic diversity per fold and stabilizes gate selection. In the cross-dataset setting, we first group ImageNet-1K into the superclasses of Table 6, partition them into $\{F_e^{\mathrm{cd}}\}_{e=1}^E$, and, guided by Table 7, assign pseudo-source and pseudo-target by matching superclass composition to each target dataset, yielding an in-ImageNet approximation of distribution shift for tuning $\tilde{\tau}$ and $\delta$ without test leakage. We de-emphasize weak-proxy alignments (e.g., DTD to *Materials & Surfaces*, EuroSAT to scene/structure proxies, UCF101 to *Sports & Recreation Gear* / *Apparel & Personal Gear* / *vehicles*) or complement them with simple appearance statistics (color histograms, texture energy, viewpoint cues), and avoid treating visually similar but nonequivalent classes as semantically identical. The estimated $\tilde{\tau}$ and $\delta$ can be found in Table 8. In practice, we have found that if we have access to unlabeled target class/domain data, this calibration process can be simplified. Through the statistics and analysis of the calibration results, we find that the $\delta$ can be set to the 5% percentile line of the target data (the logit difference between the maximum confidence prediction and the second largest). The $\tau$ can be directly set to 0, with different offsets (intercepts) on different kinds of datasets. For example, on comprehensive datasets like ImageNet, the offset is 2 or 3. On other datasets, this value is usually between -0.5$\tilde{0}$. And the corrected prediction accuracies are on par with the calibrated results.

## B.3. Pseudo-code of NeRP

We provide PyTorch-style Pseudocode to better illustrate how NeRP corrects the mispredictions. Overall, we construct a Bayes-style posterior log-odds by adding this prior term to the *observed* sample logit margin. When the prior is strong but the evidence is weak, we regard the prediction as exhibiting a prior-dominated directional bias and thus likely to be a misprediction. In this case, we perform a local decision flip within the easily confusable neighborhood to correct the

*Table 7.* Dataset-to-ImageNet-1K superclass alignment

| Dataset | Primary ImageNet-1K superclasses | Notes |
|---|---|---|
| Caltech101 | Mammals; Birds; Insects & Invertebrates; Vehicles–Air/Water/Land; Household Objects; Electronics & Imaging; Tools & Machinery; Musical Instruments & Audio; Plants–Flowers | Broad coverage; no face class in 1K. Treat portraits as weak nearest to mammals/apparel context. |
| Oxford-IIIT Pets | Mammals (incl. pets) | Strong overlap with cat/dog breeds. |
| Stanford Cars | Vehicles–Land | Coarse vehicle categories only. No fine-grained model/year in 1K. |
| Flowers102 | Plants–Flowers | Not all 102 species present. Use visually closest flower classes. |
| Food-101 | Food & Beverages | Many staples present. Complex dishes approximated by appearance. |
| FGVC Aircraft | Vehicles–Air | Fine-grained aircraft types absent. Use aircraft macro-classes. |
| SUN397 | Natural Scenes & Landforms; Buildings & Large Structures | Scene coverage is sparse in 1K. Align by landform/building appearance. |
| DTD | Materials & Surfaces (primary); Household Objects / Apparel & Personal Gear (secondary) | Use material/texture similarity on fabric/wood/stone/metal surfaces. |
| EuroSAT | Natural Scenes & Landforms; Buildings & Large Structures (proxy) | Aerial viewpoint; align by land-cover color/texture blocks and edges. |
| UCF101 | Sports & Recreation Gear; Apparel & Personal Gear; Vehicles–Water/Land (proxy) | Actions missing in 1K; use equipment/attire and vehicles as static proxies. |

*Table 8.* Calibrated Results on base-to-novel task. For each item, the first row is $\delta/\tilde{\tau}$, and the second row shows whether to use a residual prior on the text side/image side. For example, T/F indicates that only the text side residual is used.

| | Image Net | Caltech 101 | Oxford Pets | Stanford Cars | Flowers 102 | Food101 | FGVC Aircraft | SUN397 | DTD | Euro SAT | UCF101 |
|---|---|---|---|---|---|---|---|---|---|---|---|
| CoCoOp | 0.003/-4.5 | 1.0/-0.5 | 1.0/0.0 | 0.15/-3.0 | 2.0/0.0 | 0.05/-3.0 | 0.008/-3.0 | 0.1/-2.0 | 0.1/-3.0 | 1.5/1.0 | 0.06/-3.0 |
| | T/T | T/T | T/F | T/T | F/F | T/F | T/F | T/T | F/F | F/F | T/F |
| PromptSRC | 0.005/-3.0 | 0.6/-0.5 | 0.2/-2.0 | 0.2/-2.0 | 2.0/0.0 | 0.05/-3.0 | 0.004/-3.0 | 0.04/-2.0 | 0.0/-3.0 | 0.6/1.0 | 0.04/-3.0 |
| | T/T | T/T | T/F | T/T | F/F | T/F | T/F | T/T | F/F | F/F | T/F |
| MMA | 0.001/-3.0 | 0.6/0.0 | 0.02/-2.0 | 0.001/-2.0 | 2.0/0.0 | 0.02/-3.0 | 0.005/-3.0 | 0.1/-3.0 | 0.005/-3.0 | 0.6/0.1 | 0.18/-3.0 |
| | T/T | T/T | T/F | T/T | F/F | T/F | T/F | T/F | F/F | F/F | T/F |
| MMRL | 0.05/-2.0 | 0.5/0.0 | 1.0/0.5 | 0.1/0.0 | 2.0/0.0 | 0.05/-3.0 | 0.4/0.3 | 0.08/-1.0 | 1.5/0.4 | 1.5/1.0 | 2.0/0.0 |
| | T/T | T/T | T/F | T/T | F/F | T/F | T/F | T/T | F/F | F/F | T/F |

prior-driven error. In contrast, when both the prior and the evidence are weak, we deem the model genuinely unable to decide and make no adjustment. The calculation of the image/text prior can be found in Alg. 3 and Alg. 2. The decision process is shown in Alg. 4. In practice, we parallelize the implementations related to the flip decision and operate at the tensor level to reduce latency. Here we are merely demonstrating the algorithmic logic simply.

# C. More Analysis

## C.1. Visualization of Neutral-reference Images

Figure 7 visualizes the pixel-wise mean of the images used for fine-tuning across different datasets. For readability, we apply a mild contrast stretch to better reveal dataset-specific characteristics. The effect is most pronounced on FGVC-Aircraft and Stanford Cars: after averaging, most high-frequency content is removed, yet the remaining low-frequency structure still reflects the dataset tendency—the coarse silhouettes of airplanes and cars. Decisions driven by such low-frequency cues are decoupled from the true object semantics in each image, thereby exposing model bias. Ideally, when the semantics are indeterminate, the model should produce maximum-entropy (i.e., near-uniform) logits; in practice, predictions often collapse onto a few classes. NeRP calibrates the outputs precisely in these scenarios.

**Algorithm 2** Pseudocode of getting image prior.

```
# nif: neutral-ref img features [1, D]
# tf: text features [C, D]
# tf_zs: zero-shot text features [C, D]

@torch.no_grad()
def _get_image_prior():
    x = load_and_transform_mean_image()
    nif, tf, tf_zs = model(x)

    s = model.logit_scale.exp()
    b = s * (nif @ tf.t()).squeeze(0)
    b_zs = s * (nif @ tf_zs.t()).squeeze(0)

    if RESIDUAL:
        ip = b_zs - b
    else:
        ip = b_zs
```

**Algorithm 3** Pseudocode of getting text prior.

```
# ntf: neutral-ref text features [1, D]
# ntf_zs: zero-shot ntf features [1, D]
# tf: text features [C, D]

@torch.no_grad()
def _get_text_prior():
    ntf_zs, ntf = encode_neutral_template()
    ntf_zs = normalize(ntf_zs, dim=0)
    ntf    = normalize(ntf, dim=0)

    s = model.logit_scale.exp()
    b_zs = s * (tf @ ntf_zs)
    b    = s * (tf @ ntf)

    if RESIDUAL:
        tp = b_zs - b
    else:
        tp = b_zs
```

## C.2. Different Neutral Text Prompts

We compare accuracy and flip error rate (FER) across 11 datasets under different Neutral Text Prompt strategies; Figure 8 reports the detailed results. From the figure, the accuracy gap between the two strategies is negligible. While the Multiple strategy can reduce FER on a few datasets, it introduces additional computational overhead with limited accuracy gains overall. For simplicity, we therefore adopt the Single strategy (used throughout the paper).

## C.3. Different Pair Generation Strategies

To construct confusable class pairs, we first explored several naïve strategies. The most direct option is manual annotation, but this is impractical in practice due to limited expert availability and the prohibitive effort required for datasets with many categories. We then attempted to mine pairs via off-the-shelf synonym lexicons, which proved ineffective for two reasons: (i) many dataset labels are multi-word phrases that are not covered by popular synonym resources, and (ii) numerous categories are not semantically related yet are visually similar, and such cases are far from rare. We therefore adopt two strategies: (1) compute class–class similarity using CLIP's text encoder, and (2) leverage large language models (LLMs). Figure 9 compares CLIP text-prototype similarity against three locally deployed, general-purpose LLMs. CLIP text prototypes perform noticeably worse, likely because CLIP is trained on noisy web image–text pairs; the resulting textual heterogeneity and the long-tail nature of rare category semantics can induce biased similarity rankings that even degrade performance. In

**Algorithm 4** Pseudocode of NeRP's flip decision.

```
# mo: model logits [B, C]
# pair_map: graph G {class_i: [...]}
# tp, ip: text/image prior vectors [C]

def _flip_by_pairs():
    out = mo.clone()
    top1 = argmax(output, dim=1)

    for r in range(B):
        i = top1[r]
        if i not in pair_map:
            continue

        eligible = []
        for j in pair_map[i]:
            if j == i:
                continue
            score = (tp[i] - tp[j])
                + (ip[i] - ip[j])
            if score < tau:
                continue
            margin = mo[r, i] - mo[r, j]
            if margin > delta:
                continue
            eligible.append(j)

        if not eligible:
            continue

    best_j = argmax(output[r, eligible])
    out[r, best_j] = out[r, i] + eps
    return out
```

*Table 9.* Domain generalization ablation study on 4 datasets. "Text" denotes text prior only, "Image" denotes image prior only, and "T&I" denotes the use of composite prior.

| Method | Source ImageNet | Target Dataset | | | | Average |
|---|---|---|---|---|---|---|
| | | -V2 | -S | -A | -R | |
| PromptSRC | 71.27 | 64.35 | 49.55 | 50.90 | 77.80 | 60.65 |
| +Text | ● | 66.02 | 49.77 | 52.35 | 78.02 | 61.54 |
| +Image | ● | 65.29 | 49.50 | 51.46 | 77.99 | 61.06 |
| +T&I | ● | 66.87 | 49.77 | 52.77 | 78.13 | **61.89** |
| MMRL | 72.03 | 64.47 | 49.17 | 51.12 | 77.32 | 60.52 |
| +Text | ● | 66.20 | 49.15 | 53.20 | 77.65 | 61.55 |
| +Image | ● | 64.49 | 48.98 | 52.44 | 77.38 | 60.82 |
| +T&I | ● | 66.63 | 49.27 | 53.33 | 77.93 | **61.79** |

contrast, LLMs possess expert-level domain knowledge and, by producing concrete class-specific descriptions that surface visual confounders, more accurately identify visually confusable category pairs.

## C.4. Single Prior vs Composite Prior

In the domain generalization task, we compare using a single prior versus our composite prior (Table 9). Empirically, we observe a stronger bias on the text branch of CLIP, which in turn makes text-side debiasing yield larger performance gains. By contrast, the image branch sometimes exhibits little to no bias, rendering debiasing counterproductive. A plausible explanation is that the image encoder learns more fine-grained visual representations, whereas fine-grained linguistic

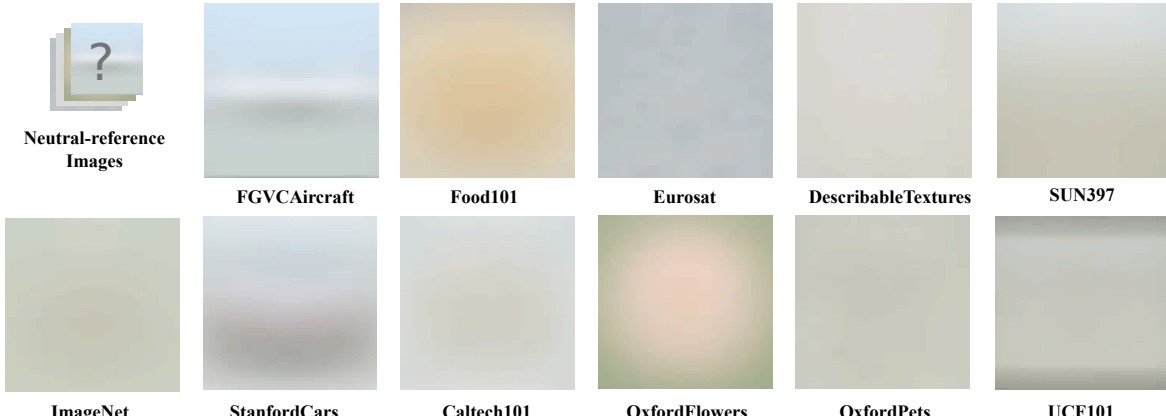

*Figure 7.* The visualization of pixel-wise average of the preprocessed training images.

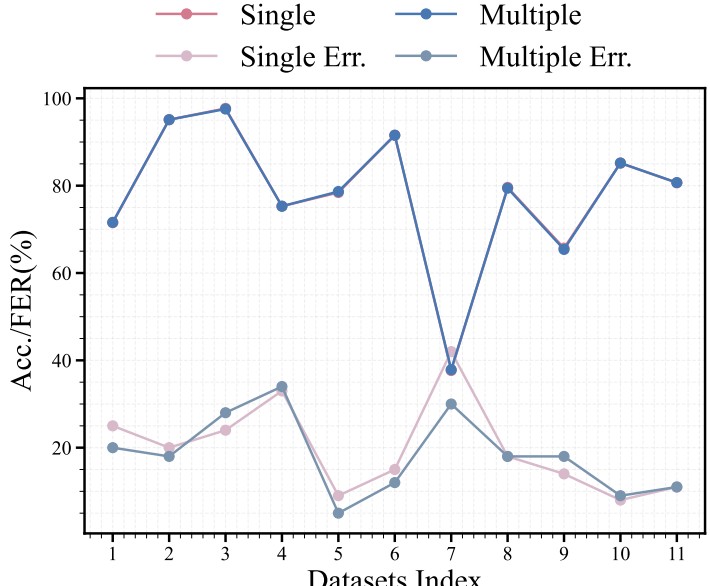

*Figure 8.* Comparison of two Neutral Text Prompt strategies across 11 datasets. Single uses one neutral prompt per dataset; Multiple uses several neutral prompts per dataset and averages the resulting features to compute the prior.

*Table 10.* Performance on different backbones.

| Method | Backbone | Base | Novel | HM |
|--------|----------|------|-------|-----|
| MaPLe | ViT-L/14 | 82.79 | 76.88 | 79.73 |
| +NeRP | ViT-L/14 | 82.79 | 77.46 | 80.03 |
| PromptSRC | ViT-L/14 | 83.24 | 76.83 | 79.91 |
| +NeRP | ViT-L/14 | 83.24 | 77.49 | 80.26 |

semantics may be underemphasized during pre-training alignment.

## C.5. Different Backbones

Beyond the default ViT-B/16 backbone used in the main paper, we also evaluate on ViT-L/14 as Table 10 shows. The trends are consistent: NeRP continues to mitigate the BNT effect.

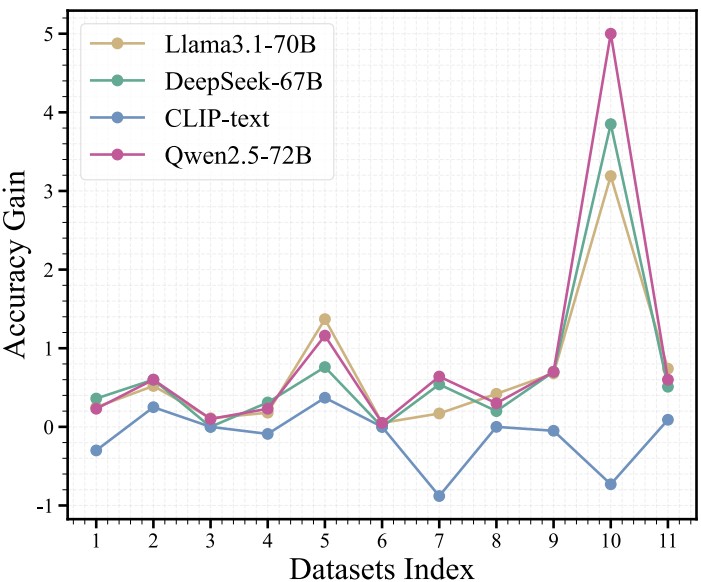

*Figure 9.* Comparison of gains across four confusable-pair generation strategies.

