# OpenReview forum: "Neutral-Reference Prompting for Vision–Language Models"
_ICML.cc/2026/Conference — ICML 2026 regular_

### Official Review · Reviewer_SptM · 2026-03-12

**Soundness:** 3
**Presentation:** 3
**Significance:** 3
**Originality:** 3
**Overall Recommendation:** 5
**Confidence:** 5

**Summary:**

This paper proposes NeRP (Neutral-Reference Prompting), a plug-and-play prompting correction strategy for efficient transfer learning of vision-language models under the Base–New Trade-off (BNT) setting. The paper argues that, beyond generic overfitting to base classes, an important source of degraded novel-class performance is asymmetric confusion between class pairs, where class A is systematically misclassified as class B but not vice versa. To address this, NeRP uses neutral text prompts and neutral-reference images to estimate class-wise prior preferences, combines them with sample-level logit margins in a Bayes-style surrogate score, and performs local correction when the prediction appears to be prior-dominated rather than evidence-driven. The method is presented as a parameter-free, post-hoc correction that can be integrated with existing prompt tuning and lightweight adaptation methods, and the paper reports improvements across few-shot and cross-domain benchmarks

**Compliance With Llm Reviewing Policy:**

Affirmed.

**Key Questions For Authors:**

Please see Weaknesses.

**Limitations:**

yes

**Strengths And Weaknesses:**

**Strengths**

1) The Base–New Trade-off is a well-known and practically relevant issue in VLM adaptation, and the paper identifies a more specific failure mode than the usual “overfitting to base classes” explanation. The focus on asymmetric confusion is interesting and could help sharpen how the community thinks about downstream bias in adapted VLMs.

2) The notion of probing model bias using a neutral-reference prompt is creative. Instead of retraining the model, the paper tries to estimate prior preference along pretrained inter-class geometry and use that information to correct prior-dominated mistakes. This is conceptually different from standard prompt tuning or adapter learning.

3) A major advantage is that NeRP is designed as a plug-and-play strategy that does not require modifying model parameters. This makes it easy to combine with diverse prompt tuning and lightweight transfer methods at inference time.

**Weaknesses**

1) The paper’s argument relies on the claim that the neutral prior gap and the expected logit margin for unseen classes tend to share the same sign with high probability, and that a neutral-reference prompt can reliably reveal the direction and magnitude of bias between confusable classes. This is an elegant idea, but also a strong modeling assumption. From the current presentation, I am not fully convinced that this assumption holds broadly across datasets, adaptation methods, and class granularities.

2) Although the paper presents NeRP as theoretically motivated, in practice the final decision rule depends on several heuristic ingredients: confusable-pair graph construction, gate parameters such as the prior and evidence thresholds, and a local flipping rule. The confusable pairs are generated using a locally deployed Qwen2.5-72B-Instruct, and calibration is based on target-emulated folds. This makes the overall framework feel less clean and less self-contained than the high-level formulation initially suggests.

3) A notable part of the pipeline depends on querying Qwen2.5-72B-Instruct to build the confusable adjacency graph. This introduces an extra source of supervision or prior knowledge outside the base VLM adaptation setup. It is unclear how sensitive NeRP is to the quality of this graph, whether simpler alternatives would work, or whether the gains partly come from this additional LLM-derived semantic knowledge rather than from the neutral-reference prompting idea itself.

---

> ### Author Rebuttal · Authors · 2026-03-30
>
> We sincerely thank the reviewer for the thoughtful and constructive comments. We especially appreciate the reviewer’s recognition of the novelty of probing bias through neutral references, as well as the plug-and-play nature of NeRP. We believe the three concerns raised here are important, and we therefore hope to provide a more precise clarification.
>
> ### Q1. The “sign consistency” claim may be too strong or too broad.
>
> ### A1.
> We appreciate this point and agree that the scope of the claim could be stated with care. Our intention is not to suggest that, uniformly across all datasets, adaptation strategies, or class granularities, the neutral prior gap necessarily tracks the expected logit margin in an unconditional sense. Rather, the theoretical result is meant to characterize the setting considered in our analysis—namely, novel classes under the low-rank deformation model together with the zero-shot stability assumption. From this perspective, we view the result as a principled explanation for why the neutral prior can be expected to provide directional information in the efficient transfer regime of interest. We will revise the wording to make this scope more explicit and to ensure that the discussion remains closely aligned with what Proposition 3.5 and Corollary 3.6 formally establish. However, we hope it is also useful to note that the empirical pattern appears to extend beyond a highly localized phenomenon. As we show in our paper, the top-20% erroneously predicted asymmetric class-confusion pairs align with the prior direction, with 79.7% overall consistency. Our evaluation spans 11 base-to-novel datasets, 10 cross-dataset transfer settings, and 4 domain-generalization benchmarks, covering fine-grained recognition tasks, as well as broader object, scene, texture, remote sensing, and human action categories. We believe these results suggest that the underlying trend is empirically observed across a relatively broad range of settings in practice.
>
> ### Q2. The final method appears somewhat heuristic and less self-contained than the high-level formulation suggests.
>
> ### A2.
> We respectfully believe that this concern partly arises from conflating the core principle of NeRP with its implementation choices. Calibration based on target-emulated folds is admittedly somewhat cumbersome, but this is done to ensure a fair comparison with other methods, namely, to ensure that the test set remains unseen during training. If some unlabeled test images are available, the calibration step can be simplified. The core principle of NeRP is actually very simple: estimate a neutral prior, compare it with the observed evidence, and intervene only in the region where the prior is dominant. The prior gate and the evidence gate are the operational realization of this principle in practice. The paper provides a risk–benefit analysis showing that, when the effective gate is properly chosen, the probability of a false flip is bounded. Moreover, the local flip itself is designed to be highly conservative: if there is no neighbor that satisfies the gating condition, the original logits remain unchanged.
>
> ### Q3. The Qwen-generated graph may introduce extra supervision, or the gains may mainly come from LLM semantic knowledge.
>
> ### A3.
> We would like to respectfully clarify that the LLM is not used as a classifier, nor does it provide sample-level supervision. It is only used offline to construct a sparse confusable neighborhood graph. The LLM is explicitly restricted to selecting candidates only from the given classes, and it never sees the test images with ground-truth labels or model predictions. Therefore, we do not believe it is accurate to describe this step as “adding supervision” in the same sense as using external labeled data. More importantly, the graph itself does not determine the final correction. It is only used to restrict the candidate neighborhood. Whether a flip occurs still depends entirely on the VLM’s own outputs. Thus, this graph is only a search-space pruning mechanism, rather than the decision maker. The reviewer also asked whether simpler alternatives could work. This comparison is already included in the appendix. We compare the confusion class pairs generated by the CLIP text encoder with those produced by several locally deployed general-purpose LLMs. The results show that using NeRP with graphs generated by the CLIP text encoder still improves performance on all datasets, and on some datasets, performs comparably to certain LLMs. Introducing an LLM can indeed help break the loop of CLIP’s internal bias. Compared with manual annotation, it only requires a one-time, fast generation for each dataset during training, with no impact at all on inference cost. Overall, it suggests that the performance gain may not be explained solely by “LLM knowledge replacing NeRP”; what is truly doing the work is still NeRP itself.

---

> > ### Author Rebuttal · Reviewer_SptM · 2026-03-31
> >
> > All concerns are addressed.

---

### Official Review · Reviewer_gcMu · 2026-03-12

**Soundness:** 4
**Presentation:** 3
**Significance:** 3
**Originality:** 3
**Overall Recommendation:** 5
**Confidence:** 3

**Summary:**

The paper investigates the Base-to-New Tunneling (BNT) problem in Vision-Language Models (VLMs). While common wisdom attributes the performance drop on unseen classes to overfitting, the authors identify asymmetric confusion—a bias inherited from imbalanced pre-training—as a primary culprit. They introduce the Neutral-Reference Prompt (NeRP), a class-agnostic anchor that reveals the direction and strength of model bias. By constructing Bayes-style posterior log-odds using these neutral priors, the model can "flip" high-confidence biased predictions on unseen classes. NeRP is proposed as a plug-and-play strategy that requires no retraining.

**Compliance With Llm Reviewing Policy:**

Affirmed.

**Final Justification:**

The authors' proposed approach has been shown to be effective on multiple datasets, including VLM, CLIP, EVA-CLIP, and SigCLIP. I recommend that the authors include these results in the paper. My final recommendation is accept.

**Key Questions For Authors:**

Have authors observed the same asymmetric confusion patterns in EVA-CLIP or SigLIP? Providing Figure 1-style visualizations for these models would significantly strengthen your claim that this is a fundamental VLM adaptation issue.

**Limitations:**

If a "plug-and-play" strategy works across the current landscape of VLMs, not just the original CLIP. If the authors can provide evidence that NeRP generalizes to EVA-CLIP and SigLIP and does not suffer from high over-correction rates, this would be better.

**Strengths And Weaknesses:**

**Strengths**
1. Shifting the focus from "overfitting" to "asymmetric confusion" caused by pre-training data imbalance is a fresh and insightful take on the BNT problem. The analysis of the "neutral prior gap" provides a strong theoretical motivation for the proposed solution.
2. As a post-hoc, plug-and-play strategy, NeRP is highly practical. Its ability to integrate with diverse existing prompt-tuning methods (e.g., CoOp, Co-CoOp) without parameter modification is a significant advantage for real-world application.
3. The use of posterior log-odds to decide when to trust evidence versus the prior is a principled way to handle model uncertainty and bias correction.

**Weaknesses**
1. It seems that only CLIP as the sole experimental backbone. While CLIP is the industry standard, modern VLM research has evolved significantly. The authors fail to demonstrate whether the "asymmetric confusion" observed in Figure 1 is a universal property of VLMs or an artifact specific to CLIP’s pre-training data. The authors should validate their findings and NeRP's effectiveness on more recent and diverse architectures, such as EVA-CLIP and SigLIP.

---

> ### Author Rebuttal · Authors · 2026-03-30
>
> We sincerely thank the reviewer for the constructive comments and the overall positive evaluation. We especially appreciate the reviewer’s key suggestion that verifying both the phenomenon of asymmetric confusion and the effectiveness of NeRP on more recent VLM families, such as EVA-CLIP and SigLIP, would make our claim much more convincing.
>
> To address this concern, we have added experiments on EVA-CLIP and SigLIP. The results show that the asymmetric confusion pattern is not unique to the original CLIP backbone: we observe the same phenomenon in both architectures. For example, split-rail fences and picket fences are visually easy to distinguish. Split-rail fences are characterized by horizontal wooden rails, wide spacing, and an open, rustic layout, whereas picket fences typically have dense vertical pickets, often with pointed tops, and a more regular residential appearance. Despite these clear structural differences, the model still tends to predict picket fence for images of split-rail fences. This suggests that the error mainly arises from a bias toward the more frequent class in the pre-training data, which is consistent with NeRP’s principle of “weak observational evidence and strong prior bias.” After introducing NeRP, we observe consistent improvements on novel classes, as shown in the tables below.
>
>
> | **EVA-CLIP**     | PromptSRC -> +NeRP |  MMRL -> +NeRP |           | PromptSRC -> +NeRP |  MMRL -> +NeRP |
> | ------------ | -----------------: | -------------: | -----------: | -----------------: | -------------: |
> | Average      |     78.43 -> 79.10 | 79.09 -> 79.83 |          DTD |     63.14 -> 63.55 | 65.00 -> 65.70 |
> | ImageNet     |     75.83 -> 76.74 | 76.25 -> 76.97 | StanfordCars |     86.42 -> 86.62 | 86.48 -> 86.58 |
> | Caltech101   |     96.49 -> 96.80 | 96.80 -> 96.90 |   Flowers102 |     79.30 -> 80.03 | 79.75 -> 80.48 |
> | Food101      |     91.66 -> 91.66 | 91.64 -> 91.69 |      EuroSAT |     81.60 -> 84.72 | 84.98 -> 87.23 |
> | FGVCAircraft |     29.72 -> 31.15 | 29.44 -> 31.81 |   OxfordPets |     97.85 -> 97.85 | 97.79 -> 97.90 |
> | SUN397       |     80.38 -> 80.50 | 80.47 -> 80.90 |       UCF101 |     80.30 -> 80.44 | 81.37 -> 81.92 |
>
>
>
> | **SigLIP**       | PromptSRC -> +NeRP |  MMRL -> +NeRP |           | PromptSRC -> +NeRP |  MMRL -> +NeRP |
> | ------------ | -----------------: | -------------: | -----------: | -----------------: | -------------: |
> | Average      |     78.92 -> 79.64 | 79.72 -> 80.44 |          DTD |     63.52 -> 64.02 | 65.56 -> 66.33 |
> | ImageNet     |     76.10 -> 76.72 | 76.68 -> 77.12 | StanfordCars |     86.38 -> 86.55 | 86.66 -> 86.83 |
> | Caltech101   |     96.07 -> 96.52 | 96.85 -> 96.96 |   Flowers102 |     79.63 -> 80.32 | 80.09 -> 80.78 |
> | Food101      |     91.91 -> 91.99 | 92.01 -> 92.06 |      EuroSAT |     81.03 -> 85.16 | 85.62 -> 87.46 |
> | FGVCAircraft |     34.29 -> 34.75 | 33.04 -> 35.58 |   OxfordPets |     97.94 -> 98.28 | 97.77 -> 98.24 |
> | SUN397       |     80.65 -> 80.87 | 80.94 -> 81.21 |       UCF101 |     80.56 -> 80.83 | 81.69 -> 82.31 |
>
>
> The reviewer also raised concerns about an excessively high flip error rate. In fact, on both EVA-CLIP and SigLIP, applying NeRP to different baseline methods leads to performance gains to varying degrees, implying that, overall, correct flips outweigh incorrect ones. We further found that, among all the flipped class pairs on ImageNet, NeRP’s flip direction matched the actual confusion direction in 88.5%/85.0% of cases for EVA-CLIP/SigLIP-based models. This is due to NeRP’s conservative design. A prediction is flipped only when two conditions are simultaneously satisfied: first, the neutral prior is strongly biased toward the currently predicted class; second, the logit-level evidence margin is small. This second condition filters out many potentially incorrect flips.
>
> Across the original CLIP and newer variants such as EVA-CLIP and SigLIP, there is no guarantee that the pre-training data are perfectly balanced. Although these models differ in training strategies, none is specifically designed to address asymmetric confusion. Therefore, we believe that the central claim of our paper should be understood as reflecting a broader property of efficient VLM transfer, rather than a CLIP-specific artifact. We hope these additional experiments strengthen the reviewer’s confidence in the generality and practical value of the proposed method.

---

> > ### Author Rebuttal · Reviewer_gcMu · 2026-04-01
> >
> > The author provides detailed experiments on EVA-CLIP and SigCLIP; all my questions have been answered, and my scores remain unchanged.

---

### Official Review · Reviewer_zp51 · 2026-03-13

**Soundness:** 3
**Presentation:** 3
**Significance:** 2
**Originality:** 3
**Overall Recommendation:** 4
**Confidence:** 3

**Summary:**

This paper studies the **base-to-novel trade-off** in efficient transfer learning for vision-language models (VLMs), where improving performance on unseen classes often harms accuracy on seen classes. The authors argue that this issue is not solely caused by overfitting, but is also related to **asymmetric confusion**, where one class is systematically misclassified as another much more often than vice versa.

To address this, the paper proposes **NeRP (Neutral-Reference Prompting)**, a plug-and-play **inference-time correction method** that does not modify model parameters. NeRP uses class-agnostic text prompts and neutral reference images to estimate class-wise prior preferences, and combines these priors with sample-level logit margins to identify predictions that are dominated by prior bias rather than image evidence. It then performs a local flip among confusable class pairs to correct such errors.

The paper also provides a theoretical analysis suggesting that these neutral priors remain informative for novel classes under a low-rank fine-tuning assumption. Experiments on base-to-novel, cross-dataset, and domain generalization benchmarks show that NeRP consistently improves novel-class performance while largely preserving base-class accuracy.

**Compliance With Llm Reviewing Policy:**

Affirmed.

**Key Questions For Authors:**

Please see the weakness.

**Strengths And Weaknesses:**

**Strengths**

1. **Clear motivation and meaningful problem formulation.**
   Instead of attributing poor novel-class generalization simply to overfitting, the paper identifies a more specific phenomenon: **asymmetric class confusion**, where samples from class A are frequently misclassified as class B but not vice versa. This perspective provides a clearer explanation of the base-to-novel trade-off (BNT) and naturally motivates a bias-correction mechanism.

2. **Simple method design that integrates well with existing approaches.**
   The proposed NeRP method does not modify model parameters and operates purely at inference time. It combines neutral priors with sample-level margins and performs local prediction flips within confusable class pairs. Importantly, the method can be easily plugged into existing approaches such as CoCoOp, PromptSRC, MMA, and MMRL.

3. **Relatively comprehensive experimental evaluation.**
   The paper evaluates the method on **11 base-to-novel benchmarks**, **10 cross-dataset benchmarks**, and **4 domain generalization benchmarks**, and reports results across several baselines. Overall, novel-class accuracy consistently improves, while base-class performance remains largely unchanged. The consistent improvements across different tasks make the approach practically appealing for an inference-time correction method. The additional analyses on calibration granularity, fold number, and confusable pair selection further support the validity of the experimental results.


**Weaknesses**

1. **The causal interpretation of the core claim is limited.**
   The paper claims that neutral priors reliably capture the direction of pretraining bias and that this bias is the main cause of novel-class misclassification. However, the current evidence mainly shows **correlation rather than causation**. Even if the prior direction aligns with the error direction for many class pairs, it does not necessarily imply that model errors are primarily driven by such priors, or that the flipping operation is truly correcting bias rather than performing a heuristic margin-based post-processing.

2. **Construction of confusable pairs introduces additional external dependency.**
   The confusable class pairs are generated using a locally deployed **Qwen2.5-72B-Instruct** model. This design is not fully responsible as part of the performance gain may come from the **LLM-provided semantic adjacency prior**. And the generated class-pair graph may vary depending on the LLM, prompts, or decoding settings.

3. **Some terminology can be improved**
   The term *“Bayes-style surrogate”* appears to resemble more of a **logit-space heuristic decomposition** rather than a strictly defined Bayesian formulation.

Overall, although the method relies on relatively strong assumptions, its empirical results are good and it consistently improves performance across different prompt-tuning architectures. At this stage, I am leaning toward acceptance.

---

> ### Author Rebuttal · Authors · 2026-03-30
>
> We sincerely thank the reviewer for the careful reading, the constructive comments, and the overall positive assessment. We are greatly encouraged that the reviewer finds our problem formulation meaningful, our method practically appealing, and our empirical evaluation relatively comprehensive. Below, we clarify the concerns.
>
> ### Q1. On the causal interpretation of our core claim.
>
> ### A1.
> We appreciate the reviewer’s point and agree that the distinction here is worth making more carefully. Our intention is not to claim a formal causal identification result in the strict sense. Rather, the theoretical claim we make is more limited in scope: under the low-rank deformation and zero-shot stability assumptions, the neutral prior gap can reflect the direction of pretraining-induced preference on novel classes, and this information can be practically useful for improving prediction decisions. In this sense, our analysis is aimed at establishing sign consistency and predictive utility within the setting considered, rather than attributing all novel-class errors to a single source.
>
> We also agree that this aspect of the presentation can be clarified further to avoid any unintended overstatement. In the current manuscript, our intention was not to suggest that overfitting is unimportant. Rather, we mean to convey that poor performance on novel classes may not be fully accounted for by overfitting alone, and that asymmetric confusion can serve as an additional mechanism that is helpful for understanding the observed behavior. To make this point maximally clear, we are very willing to further soften the wording in the revision.
>
> ### Q2. On the external dependency introduced by Qwen-generated confusable class pairs.
>
> ### A2.
> We would like to respectfully clarify that the performance gains of NeRP do not fundamentally come from using confusable class pairs generated by Qwen. In the Appendix Fig.8, we compare the confusion class pairs generated by the CLIP text encoder with those produced by several locally deployed general-purpose LLMs. The results show that NeRP with graphs generated by the CLIP text encoder still improves performance on all datasets, and on some datasets performs comparably to LLMs. This suggests that the effectiveness mainly comes from NeRP itself. LLMs simply provide external priors that help CLIP break the loop of its own internal bias, leading to further improvement. Compared with manual annotation, LLM-based generation of confusion pairs greatly reduces the workload, requiring only a one-time, fast generation for each dataset during training, with no effect on inference cost. The graph does vary with the LLM, prompts, and decoding settings. However, NeRP itself makes conservative and reliable flips. As discussed above, the appendix shows that NeRP’s gains are robust across different confusion-pair generation strategies. Our goal is simply to pair NeRP with a more suitable strategy to achieve larger gains, which does not affect its core innovation.
>
> ### Q3. On the term “Bayes-style surrogate.”
>
> ### A3.
> We appreciate this suggestion and agree that the terminology can be made more precise. We do not claim that our score is a strictly defined Bayesian posterior. This is exactly why we use qualifiers such as “Bayes-style” or “Bayes-inspired” in the paper. More precisely, our formulation is a Bayes-inspired logit-space surrogate: under the von Mises–Fisher model, the observed sample margin can be approximately interpreted as a log-likelihood ratio, while the neutral prior gap acts as a prior-like offset. In the revision, we will refine the wording accordingly and make this approximate nature more explicit, so as to avoid any impression that we are claiming an exact Bayesian derivation.

---

> > ### Author Rebuttal · Reviewer_zp51 · 2026-04-03
> >
> > My concern has been addressed, and I will maintain my original rating.

---

### Official Review · Reviewer_iouB · 2026-03-13

**Soundness:** 3
**Presentation:** 3
**Significance:** 3
**Originality:** 3
**Overall Recommendation:** 4
**Confidence:** 4

**Summary:**

The paper investigates the prevalent base-new trade-off in Vision-Language Models (VLMs), positing that the degradation of known-class accuracy upon improving unseen-class performance is driven by asymmetric confusion biases inherent in imbalanced pre-training data, rather than mere overfitting. As a remedy, it presents Neutral-Reference Prompting (NeRP), a training-free inference strategy. NeRP quantifies class-wise prior biases using neutral prompts and reference images, subsequently deriving Bayes-style posterior scores by fusing these priors with sample likelihoods. Furthermore, the framework incorporates a local decision-flip mechanism guided by LLM-generated confusion graphs to rectify predictions when prior bias outweighs empirical evidence. Comprehensive evaluations across 15 few-shot and cross-domain benchmarks confirm the efficacy of NeRP.

**Compliance With Llm Reviewing Policy:**

Affirmed.

**Final Justification:**

Thank you to the authors for the detailed rebuttal, which addresses my concerns effectively. I believe incorporating the above disccusion into the final version will strengthen the paper. I recommend acceptance.

**Key Questions For Authors:**

See weakness.

**Limitations:**

yes

**Strengths And Weaknesses:**

**Strengths**
- Well-motivated problem: The paper identifies a genuine and important issue in vision-language models—that CLIP's predictions are influenced by dataset-specific biases and imbalanced pretraining distributions. Recognizing that these biases persist even in few-shot and zero-shot scenarios, and proposing to address them through principled bias correction rather than simply fine-tuning, represents a valuable research

- Good performance. NeRP achieves competitive results on multiple benchmark datasets (ImageNet, StanfordCars, FGVCAircraft, etc.) under various settings including base-to-novel generalization, few-shot learning, and zero-shot classification.

- The paper is very well written, with clear motivations, sufficient technical explanations and illustrative visualizations.


**Weakness**

- The paper uses mean images or generic text prompts as "neutral references" to estimate class prior biases, but this assumption lacks rigorous justification. Mean images inevitably retain statistical biases from the training set—for instance, in StanfordCars, if training samples are predominantly side-view sedans, the mean image preserves "side-view angles," "horizontal body lines," and "road backgrounds," causing queries to favor sedan classes over trucks or SUVs, reflecting training imbalance rather than true neutrality. Similarly, generic text like "a photo of a texture" occupies a specific semantic position in CLIP's embedding space, naturally closer to material-related classes than abstract concepts, and different classes have varying template compatibility ("a photo of a sedan" is common while "a photo of aerodynamic forces" is rare). More fundamentally, CLIP as a discriminative model produces similarity scores reflecting geometric relationships in embedding space, not Bayesian prior probabilities—the paper equates these without theoretical justification. Critical experiments are missing: systematic comparisons of different neutral reference variants (Gaussian noise, uniform gray images, alternative text templates like "an image," "something," "a photo," "an object") to verify prior estimation's sensitivity and stability.

- Circular dependency in confusion graph construction: Using LLM to generate confusion graphs for guiding decision flipping presents a logical problem—if the LLM can accurately predict which classes will be confused, why not directly use the LLM's knowledge to improve classification? The paper does not verify the consistency between LLM-generated confusion graphs and model confusion matrices, nor does it compare the effectiveness of manually annotated confusion graphs.

- Hallucination risks in LLM-generated confusion graphs: The method relies on LLMs to generate confusion graphs that determine which classes are easily confused. LLM's common sense may not align with specific visual dataset distributions (for example, in medical imaging or specialized texture datasets, LLMs may fail to predict which classes are visually similar). This introduction of non-visual priors may inject new noise.

- Insufficient evaluation of computational overhead: The paper claims to be a "plug-and-play" inference-time method, but it involves additional steps such as neutral prompt computation, reference image inference, and LLM-based confusion graph generation—how much does actual inference time increase? What is the scalability in large-scale datasets? The paper only shows accuracy improvements without reporting inference speed and memory consumption.

- Missing comparisons with other bias correction methods: The paper primarily compares with CLIP and CoOp-type prompt learning methods, but lacks comparisons with post-processing correction methods (such as C-TPT[1], O-TPT [2], etc.)—these methods can similarly adjust prediction distributions at inference time. Does NeRP's advantage solely come from Bayesian formalization?

- Missing Related Work. The "Related Work" section lacks comparisons with some of the latest research [3,4,5,6] on prompt learning.

[1] C-tpt: Calibrated test-time prompt tuning for vision-language models via text feature dispersion. ICLR 2024

[2] O-TPT: Orthogonality Constraints for Calibrating Test-time Prompt Tuning in Vision-Language Models. CVPR 2025

[3] DPC: Dual-Prompt Collaboration for Tuning Vision-Language Models.  CVPR 2025

[4] Rethinking Few-Shot Adaptation of Vision-Language Models in Two Stages.  CVPR 2025

[5] Hierarchical Cross-modal Prompt Learning for Vision-Language Models. ICCV 2025.

[6] VaMP: Variational Multi-Modal Prompt Learning for Vision-Language Models. NeurIPS 2025.

---

> ### Author Rebuttal · Authors · 2026-03-30
>
> We sincerely thank the reviewer for the thoughtful comments and recognition of this work as a valuable contribution.
> ### Q1. Neutral prompts are not neutral & sensitivity analysis
> ### A1.
> This concern may be based on a misunderstanding. In the appendix, we already visualize the mean image for each dataset. These images do **not** preserve instance-level semantics such as “horizontal car-body lines” or “road background.” Instead, they completely remove semantic content and retain only low-frequency statistics. The same **misunderstanding** applies to the text side. We do **not** use one universal neutral text such as “a photo of a texture” across all datasets. Instead, we use dataset-specific neutral prompts. Also, we do not claim that CLIP similarities are exact Bayesian priors. The paper explicitly uses the term **“Bayes-style surrogate”** and only states that the sample margin can be approximately interpreted as a log-likelihood ratio under a von Mises–Fisher model.
>
> The claim that a “critical experiment is missing” may be unfair. The reviewer’s suggested uniform gray images and Gaussian noise **lie off the model’s data manifold**; such invalid references can only probe the model’s response to OOD inputs rather than the bias, and therefore contradict our theoretical setup. Also, in Appendix Fig.8, we have already compared the effects of single versus multiple neutral prompt templates to verify prior estimation's sensitivity and stability.
>
> ### Q2. If the LLM knows which classes are confusable, why not directly use the LLM to improve classification?
> ### A2.
> We believe this concern may stem from a misunderstanding. The LLM does **not** predict the correct class, **nor** does it decide the final flip. It only provides **loose candidate neighbors**, while the final decision is made entirely by the model’s own outputs: the neutral prior gap and the observed evidence margin. For datasets in our paper, the LLM has enough commonsense knowledge to tell which classes may plausibly be confused. NeRP uses this only as a **search-space restriction**. Even if the LLM proposes a wrong pair of visually dissimilar classes, the evidence shown by the current VLM's prediction will usually be very strong, thus NeRP will **not** flip the decision.
>
> ### Q3. LLMs can introduce noise(e.g., on texture data).
> ### A3.
> We respectfully clarify that NeRP achieves consistent performance gains on the DTD (a **texture dataset** used in our paper). Although LLMs may introduce hallucinations and noise, NeRP is conservative and fault-tolerant. For example, we allow up to 10 similar neighbors for each class, and NeRP selects the correct flipping target based on a neutral prior and sample evidence by itself, rather than relying on the LLM. We also use detailed prompts (see Appendix) to mitigate LLM hallucinations as much as possible.
>
>
> ### Q4. The paper does not evaluate computational overhead & compare against TPT methods.
> ### A4.
> We believe that the comparisons against test-time prompt tuning (TPT) methods may not be directly applicable.
> - TPT and our method address **different settings**. TPT adapts prompts using unlabeled test data, whereas our method maintains strict **invisibility of test data**.
> - Some work like Self-TPT and O-TPT has shown that simply stacking prompt learning and TPT does **not always lead to better results**. It can even hurt performance for methods like PromptSRC that include regularization constraints. How to effectively combine them is beyond the scope of this paper.
> - TPT-style methods incur **much larger inference overhead**. Their extra cost is typically **hundreds of milliseconds per image**, since they require online prompt updates at test time. The construction of our confusion graph is completed **before inference**, so the inference overhead is minimal through tensor parallel operations (extra mem overhead is only KB-level).
> ### Inference-time comparison
> (ViT-B/16 scale, RTX A6000, ImageNet settings)
>
> |Method|Cost|
> |- |- |
> |TPT|412.5 ms/**img**|
> |C-TPT|447.4 ms/img|
> |O-TPT|480.9 ms/img|
> |NeRP extra latency||
> |CoOp inference|11.2 ms/img, 31.4 ms/batch (bs=32)|
> |+NeRP|+0.3 ms/**batch** (bs=32)|
> |+NeRP|+0.4 ms/batch (bs=64)|
> |+NeRP|+0.7 ms/batch (bs=128)|
>
> - Our method works very differently from TPT and addresses a different problem, so the two are not mutually exclusive. Their potential synergy can be explored in future work.
>
> ### Q5. The related work section misses several papers.
>
> ### A5.
>
> We would like to clarify that **DPC[3] is already discussed and cited in our Related Work section.** For HicroPL[5], we attempted to reproduce the method before submission but observed results that differed substantially from those reported in the original paper. We therefore chose not to include it as a main comparison. For VaMP[6], the code was still unavailable when we completed our paper. The works mentioned could be cited in the revision for more detailed discussion.

---

> > ### Author Rebuttal · Reviewer_iouB · 2026-04-01
> >
> > Thank you to the authors for the detailed rebuttal, which addresses my concerns effectively. I believe incorporating the above disccusion into the final version will strengthen the paper. I recommend acceptance.

---

### Decision · Program_Chairs · 2026-04-30

**Decision:**

Accept (regular)

**Comment:**

The paper proposes a neural prompting method for debiasing VLM predictions during transfer learning. It received positive assessments from the reviewers, including two weak accepts and two accepts. The reviewers agreed that the paper is well motivated by intriguing observations, and that the training-free prompt-tuning approach is both novel and thoroughly validated. Some concerns were raised, including computational overhead, the justification for constructing confusion graphs using LLMs, and the need for additional validation across more CLIP variants. The authors successfully addressed these concerns during the rebuttal period.

The AC considers debiasing through neural prompting for VLMs to be a novel contribution to the field. The experimental results are thorough and convincing. Therefore, the AC recommends acceptance. The AC encourages the authors to revise the paper based on the reviewers’ feedback.